# Interactions between strains govern the eco-evolutionary dynamics of microbial communities

Akshit Goyal[1], Leonora S Bittleston[2,3], Gabriel E Leventhal[2], Lu Lu[2], Otto X Cordero[2]*

[1]Physics of Living Systems, Department of Physics, Massachusetts Institute of Technology, Cambridge, United States; [2]Department of Civil and Environmental Engineering, Massachusetts Institute of Technology, Cambridge, United States; [3]Department of Biological Sciences, Boise State University, Boise, United States

**Abstract** Genomic data has revealed that genotypic variants of the same species, that is, strains, coexist and are abundant in natural microbial communities. However, it is not clear if strains are ecologically equivalent, and at what characteristic genetic distance they might exhibit distinct interactions and dynamics. Here, we address this problem by tracking 10 taxonomically diverse microbial communities from the pitcher plant *Sarracenia purpurea* in the laboratory for more than 300 generations. Using metagenomic sequencing, we reconstruct their dynamics over time and across scales, from distant phyla to closely related genotypes. We find that most strains are not ecologically equivalent and exhibit distinct dynamical patterns, often being significantly more correlated with strains from another species than their own. Although even a single mutation can affect laboratory strains, on average, natural strains typically decouple in their dynamics beyond a genetic distance of 100 base pairs. Using mathematical consumer-resource models, we show that these taxonomic patterns emerge naturally from ecological interactions between community members, but only if the interactions are coarse-grained at the level of strains, not species. Finally, by analyzing genomic differences between strains, we identify major functional hubs such as transporters, regulators, and carbohydrate-catabolizing enzymes, which might be the basis for strain-specific interactions. Our work suggests that fine-scale genetic differences in natural communities could be created and stabilized via the rapid diversification of ecological interactions between strains.

*For correspondence: ottox@mit.edu

Competing interest: The authors declare that no competing interests exist.

## Editor's evaluation

How easily is one species replaced by another system in an ecosystem, and what does it take so that two species are no longer equivalent? This is a central issue of ecology, which has been addressed in this elegant study. The rule of thumb the authors come up with, that genetic differences between two bacterial strains greater than about 100 bp are a good predictor of these strains being no longer ecologically equivalent, is likely to be one that will be highly cited in future.

## Introduction

In nature, microbial communities contain individuals on a continuum of phylogenetic diversity, where both evolutionarily distant and proximate members coexist (*Ding et al., 2016*; *Fierer et al., 2007*). Members in the same communities can belong to different domains of life, such as archaea, bacteria, and fungi, and at the same time, can stably exist with extremely closely related relatives, a few single-nucleotide polymorphisms (SNPs) apart (*Schloissnig et al., 2013*; *Han et al., 2014*; *Faith et al., 2013*;

*Kashtan et al., 2014*). Collectively, closely and distantly related community members perform many vital ecological functions such as the regulation of biogeochemical cycles and fiber digestion in animal guts (*Rousk and Bengtson, 2014*; *David et al., 2014*). However, identifying how these different levels of diversity interact and co-evolve in complex communities remains an elusive and challenging problem.

A common strategy to study this problem is to analyze the dynamics of complex communities at different levels of diversity with metagenomic sequence data. Such data have been successfully leveraged to reconstruct the linkage between polymorphic sites in natural microbial populations, effectively allowing one to resolve both species and strain abundances (*Good et al., 2017*; *Garud et al., 2019*; *Plucain et al., 2014*; *Frenkel et al., 2015*). With these data, we can ask the important question: which level of diversity—strains, species, genera, families (or even broader taxonomic units)—have the strongest interactions and influence on community dynamics? In other words, what is the appropriate level of coarse-graining to describe and predict microbial community dynamics? To illustrate, if interactions at the species level are conserved (similar) at lower levels (like strains), but not at higher levels (like genera or families), then a species-level description would be appropriate. However, if strains within species display distinct dynamics and interactions with strains of other species, then a coarse-grained, species-level description would not be appropriate to describe long-term dynamics. This question is hard to answer using natural systems such as mammalian guts since they are often far from equilibrium, where extrinsic factors like host control can be much stronger drivers of community dynamics than interactions between the community members themselves (*Spor et al., 2011*). By domesticating natural communities in controlled laboratory conditions, however, we can study community dynamics near equilibrium (*Frazão et al., 2019*). In such domesticated communities, we expect changes in community composition to be induced primarily by intrinsic factors such as biological interactions. Further, by studying multiple communities domesticated in parallel in the same abiotic environment, we can disentangle which observations about community dynamics are repeatable and general, and which ones are chance and context-specific.

Here, we show that strains—which are genetic variants of the same species—exhibit the strongest dynamical correlations across all taxonomic levels in microbial communities. These results stem from tracking 10 domesticated microbial communities from the pitcher plant *Sarracenia purpurea* for more than 300 generations. By analyzing dynamics at several phylogenetic levels, we show that community composition varies most strongly at the level of strains. Further, interactions within each community were strain-specific: at times decoupling the dynamics of one strain from its close relative. Remarkably, as few as 100 base pair differences across strain genomes (~99.99% similarity) were sufficient for strain dynamics to diverge from each other. Finally, we show that strains can differentiate by fine-tuning only a handful of functional categories in their genomes, such as transcriptional regulators, metabolite transporters, and tricarboxylic acid (TCA) cycle enzymes. Together, our results highlight that while broader taxonomic units like families may help predict community dynamics at short timescales, strains may be the relevant unit of taxonomic coarse-graining at which to study interactions and dynamics in microbiomes at longer timescales, not merely a descriptive detail.

## Results

### Naturally occurring strains coexist for hundreds of generations in laboratory microcosms

We followed the eco-evolutionary dynamics of 10 replicate microbial communities derived from the carnivorous pitcher plant *S. purpurea* (*Figure 1a*). We began by sampling communities from 10 distinct pitchers from plants belonging to the same bog, and after sampling, filtered out particles larger than 3 μm (to focus on bacteria). We then transferred and propagated each ecological replicate community separately through serial passaging in a medium consisting solely of acidified water and ground cricket powder as the nutrient source (see Materials and methods). This medium mimics, in part, the ecological conditions of the pitchers from which we derived these microbes.

To stably maintain high diversity, we initially propagated communities at a low dilution factor (1:2) every 3 days for 21 transfers, during which they reached distinct, rich, and stable equilibria (*Figure 1a*; ecological dynamics studied in a previous publication [*Bittleston et al., 2020*]). To study community dynamics over evolutionary timescales, we followed them using a similar protocol, but at a higher

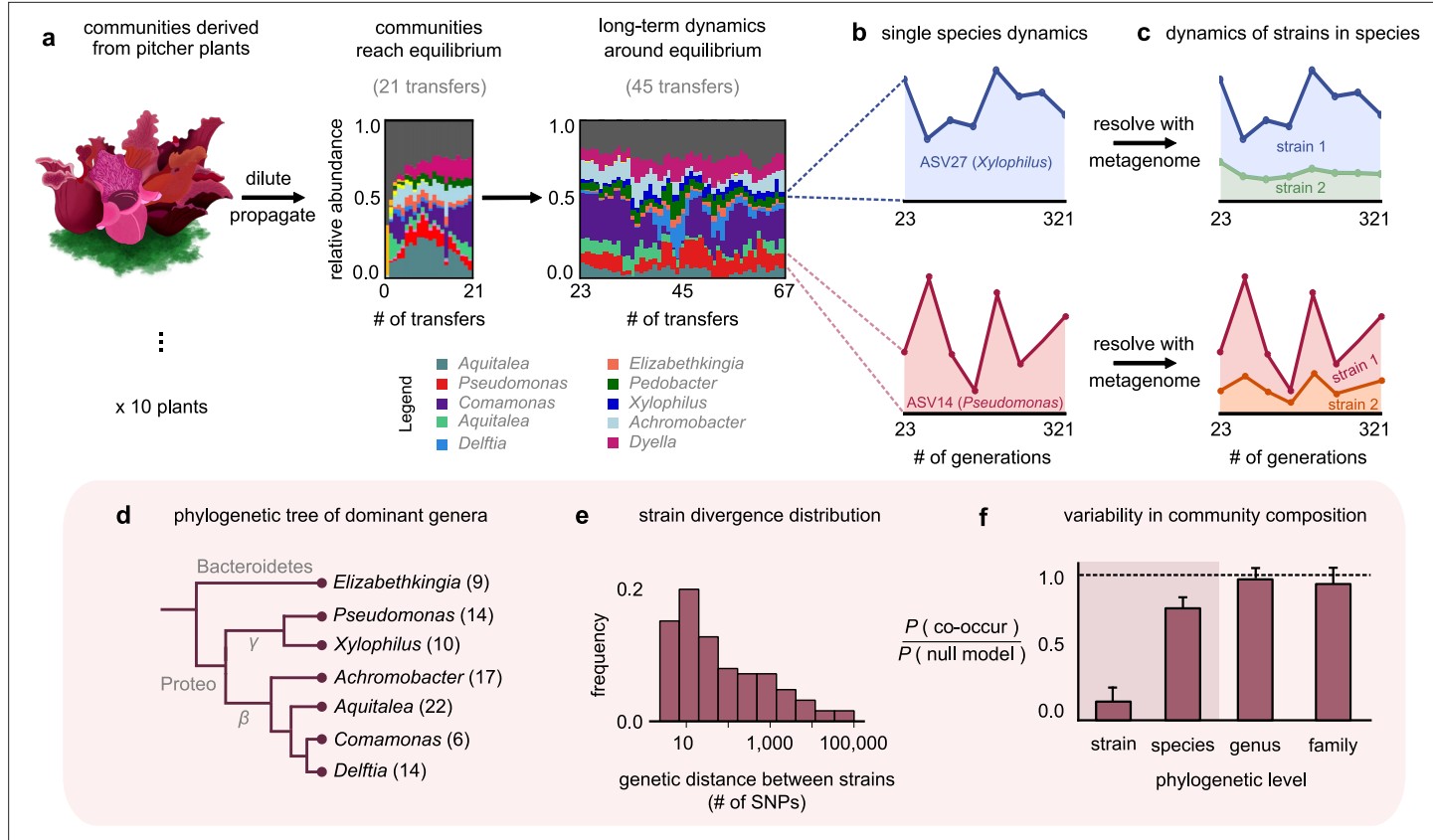

**Figure 1.** Closely related strains coexist for hundreds of generations in pitcher plant-derived microbial communities. (**a**) Diagram illustrating our experimental protocol. Stacked bar plots show the composition of one community (M06) at the amplicon sequence variant (ASV) (species) level sampled at each transfer; each color corresponds to a unique ASV that we tracked further using metagenomic sequencing, with their genera in the legend. (Illustration credit: Michelle Oraa Ali.) (**b**) Relative abundances of ASV27 (blue) and ASV14 (pink) with the approximate number of generations (see Materials and methods); the eight time points shown correspond to those for which we collected metagenomic data. (**c**) Relative abundances of strains identified in ASV27 and ASV14 using metagenomes. The shaded colors correspond to the abundance of each of the two strains. (**d**) Phylogenetic tree of the dominant strain-containing taxa across all 10 communities, with the identified genus names labeled. Brackets indicate the number of detected strains belonging to each genus. (**e**) Distribution of the genetic distance (divergence) between strains belonging to the same species, measured in the number of detected single-nucleotide polymorphisms (SNPs) differentiating them. (**f**) Bar plot showing the probability with which two members of the same taxonomic group (family, genus, species, etc.) co-occurred in a sample, normalized with a null model where all members were distributed randomly across communities. Dashed line indicates the null expectation.

The online version of this article includes the following figure supplement(s) for figure 1:

**Figure supplement 1.** Long-term species dynamics of all 10 experimental microbial communities.

**Figure supplement 2.** Non-metric multidimensional scaling (NMDS) plot of community compositions at the species level.

**Figure supplement 3.** Most (>99%) single-nucleotide polymorphisms (SNPs) are biallelic (have two alleles).

**Figure supplement 4.** Single-nucleotide polymorphism (SNP) trajectories within a species are highly correlated.

**Figure supplement 5.** Single-nucleotide polymorphisms (SNPs) within strains tightly cluster together.

**Figure supplement 6.** Single-nucleotide polymorphism (SNP) clusters are robust to alternate clustering methods.

**Figure supplement 7.** More examples of single-nucleotide polymorphism (SNP) clusters.

**Figure supplement 8.** Changes in strain relative frequencies.

**Figure supplement 9.** Changes in strain frequencies often influence their overall species abundances.

**Figure supplement 10.** Distribution of correlations between species' relative abundances inferred using read mapping and 16S rRNA sequencing.

dilution factor (1:100) and less frequent transfer rate (every 7 days) for more than 300 generations (46 additional transfers). An increased dilution factor allowed a higher number of generations to pass between successive passages, and a weekly transfer rate made community propagation experimentally manageable over a full year. To resolve both coarse- (species) and fine (genotype)-scale

dynamics, we performed 16S rRNA and deep metagenomic sequencing, respectively. Specifically, we used 16S rRNA sequencing to follow species composition (with species defined as having identical 16S rRNA); this gave us the 'ecological' dynamics of each community. Metagenomic sequencing, which we performed at eight evenly spaced time points between transfers 23 and 67, allowed us to follow 'evolutionary,' or genotype-level, changes in each community (*Figure 1c*). Together, this setup enabled us to track the eco-evolutionary dynamics of all 10 communities.

At the species level (variants with the same 16S rRNA sequence), the composition of each community fluctuated dynamically, but remained around the same equilibrium state for >300 generations (*Figure 1a*; median ~31% temporal coefficient of variation in the abundance of a species, *Figure 1—figure supplement 1*; and tightly clustered low-dimensional dynamics, *Figure 1—figure supplement 2*). We hypothesized that the genetic variation occurring during the experiment, through mutations and recombination, might be responsible for these fluctuations. To test this, we mapped metagenomic reads from each sample to a database of 33 reference genomes belonging to isolates from our communities (see Materials and methods). To avoid sequencing-related artifacts, we ensured that read mapping was competitive, that is, we only used those sequenced reads that mapped unambiguously to one genome. Importantly, because our reference genomes belonged to isolates from our communities, any detected genetic variation indicated the presence of variants relative to a resident of these communities.

Most (97%) genetic changes were in the form of SNPs, an overwhelming majority (~98%) of which were detected in the first sequenced time point (~23 generations). This suggested that the communities had significant preexisting, or standing, genetic variation. Given the large degree of divergence (~1% genome-wide divergence for some pairs; *Figure 1e*) and estimated divergence time based on bacterial mutation rates (~100 generations to diverge by one SNP; see Materials and methods), the preexisting variation likely came from genetic variants in the plants rather than from variants arising during the first 23 generations of lab propagation. Interestingly, the large divergence times of the genetic variants (~100 s of years) were much larger than the age of the pitchers we sampled (~3 months; see Materials and methods). This suggested that the preexisting variants represented different isolates (or strains) that had arrived in the pitchers through separate colonization events rather than having diversified within a pitcher during its lifetime. Therefore, we rejected our original hypothesis that mutations and recombination occurring during the experiment were the main driver of community dynamics. Instead, we focused on the dynamics of preexisting variants. Almost all (~99%) SNPs were biallelic (had only two variants; *Figure 1—figure supplement 3*), which allowed us to track variant dynamics in terms of the temporal trajectory of each SNP.

The temporal trajectories of SNPs in the same reference genome (species) were highly correlated, with their allele frequencies increasing and decreasing together (mean correlation coefficient 0.8; p<0.01; *Figure 1—figure supplement 4*). Such highly correlated allele frequency trajectories are hallmarks of genetic linkage (*Good et al., 2017*; *Roodgar et al., 2009*) and suggest that the large number of SNPs in each species were co-localized in a small number of genotypes or strains. We clustered the allele frequency trajectories and could statistically detect at most two strains for each species (see Materials and methods and *Figure 1—figure supplement 5*). For each cluster (strain), we estimated its frequency within the whole community as the fraction of the species' abundance that corresponded to the strain (see Materials and methods). We calculated the relative abundance of each strain by multiplying its frequency with the relative abundance of the species it belonged to. Together, we concluded that within each community there was a pair of strains underlying most taxa, including the phylogenetically diverse *Elizabethkingia*, *Aquitalea*, and *Delftia* (*Figure 1d*). Within each species, strain dynamics displayed vastly different patterns. For some species, only one of the strains changed appreciably in abundance over the experiment (*Figure 1c*, top; *Xylophilus* sp.). For other species, both strains fluctuated constantly throughout the experiment (*Figure 1c*, bottom; *Pseudomonas* sp.). From an evolutionary standpoint, we observed a roughly bimodal distribution of the overall change in strain frequencies across different species in our communities (*Figure 1—figure supplement 8*, *Figure 1—figure supplement 9*). In some cases, the abundance of a species stayed relatively constant while the strains comprising it fluctuated relatively, similar to what has been observed in other systems such as the human gut (*Garud et al., 2019*; *Roodgar et al., 2009*; *Zhao et al., 2019*).

Since we observed different conspecific strains in different communities, we next asked at which phylogenetic level the 10 replicate communities were most variable at, that is, strains, species, genera,

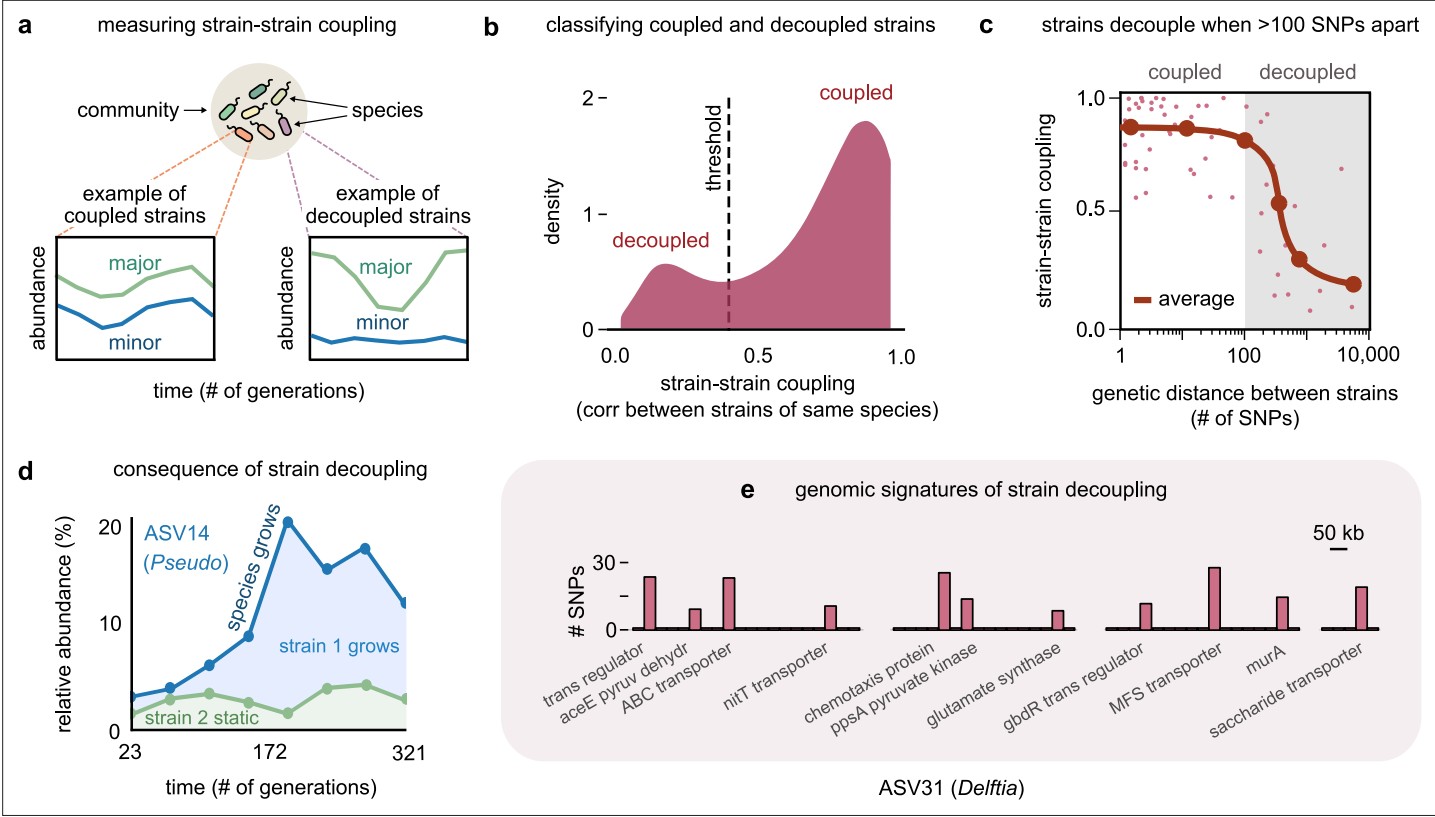

**Figure 2.** Even highly related strains (~100 single-nucleotide polymorphisms [SNPs] apart) can decouple in their dynamics. (**a**) Schematic showing examples of strain–strain coupling. We defined strain–strain coupling as the temporal correlation between strain abundances belonging to the same species in a community; coupled strains (left) had highly correlated abundances while decoupled strains (right) were uncorrelated. (**b**) Distribution of the strain–strain coupling across all species and communities, smoothed using Gaussian kernel density estimation; dashed line shows the threshold coupling used to classify strains as coupled and decoupled (see Materials and methods). (**c**) Strain–strain coupling as a function of the genetic distance between strains. Each gray point represents a conspecific strain pair. The solid red line shows a moving average (LOESS fit). (**d**) Relative abundance of the species labeled ASV14 of the genus *Pseudomonas* in community M03, and its underlying strains over time; the blue shaded region represents strain 1 while the green region represents strain 2. The solid blue line shows the total species abundance. (**e**) Bar plot showing the number and genetic location of SNPs in the core genome of ASV31 from community M04, whose strains were decoupled and differed by 186 SNPs. Each bar shows the number of SNPs in one gene along with its annotation. Only SNPs belonging to annotated genes are shown.

The online version of this article includes the following figure supplement(s) for figure 2:

**Figure supplement 1.** Null distribution of strain–strain coupling from a consumer-resource model with no differences between conspecific strains.

**Figure supplement 2.** Strain–strain coupling distribution is robust to using an alternate measure.

**Figure supplement 3.** Distribution of strain–strain coupling with the sign of the correlation.

or families. For this, we calculated the probability that two members of the same taxonomic group (family, genus, species, etc.) co-occurred in a sample, normalized against a null model where all members were distributed randomly across communities (see Materials and methods). We found that community composition across all 10 replicates was much more variable at the level of strains than species (*Figure 1f*). This result is consistent with previous work that natural microbial communities are taxonomically variable at finer phylogenetic levels (strains and species), but more similar at coarser levels (families) (*Goldford et al., 2018*; *Louca et al., 2016*). Together, we concluded that natural strains of the same species can coexist in microbial communities over hundreds of generations, maintaining the communities in distinct stable states.

## Highly related strains can decouple in their dynamics

Motivated by the observation that closely related strains persisted in communities, we asked whether the dynamics of each strain were similar, or coupled, to that of their closest relatives. To answer this question, we measured a strain–strain coupling for strains of the same species, defined as the

correlation between their abundance trajectories (*Figure 2a* and Materials and methods). Strain dynamics could either be highly correlated ('coupled strains,' *Figure 2a*) or be uncorrelated ('decoupled strains'). Measuring strain–strain coupling for each species across all 10 communities revealed a bimodal distribution, whose two modes were occupied by highly correlated and uncorrelated strain pairs, respectively (*Figure 2b*). The shape of this distribution allowed us to reliably classify strain pairs as either decoupled (coupling <0.4, the inflection point of the distribution; see Materials and methods) or coupled (coupling >0.4). A null model with no ecological differences between conspecific strains instead showed a unimodal distribution of coupling (mean coupling 0.95, *Figure 2—figure supplement 1* and Materials and methods). This control suggested that the observed incidences of strain decoupling were much higher than expected simply by chance ($p < 10^{-3}$; Kolmogorov–Smirnov test). We thus concluded that ~20% of conspecific strains, belonging to the same species, displayed rather dissimilar or decoupled dynamics.

To test if the genetic distance between strains influenced their strain–strain coupling, we plotted the average coupling as a function of the genetic distance between a conspecific strain pair (see Materials and methods). Remarkably, the average strain–strain coupling decreased sharply beyond a genetic distance of just about 100 SNPs (*Figure 2c*). This suggested that changes in as few as 100 bps, corresponding to roughly 0.01% of these genomes, were sufficient to decouple strain dynamics. A consequence of such decoupling was that in species with decoupled strains drastic changes in the species' abundance were driven only by changes in one of the strains (*Figure 1c* and *Figure 2d*, top).

To understand the genomic signatures of strain decoupling, we examined the location and putative function of SNPs in a pair of highly related but decoupled strains (see Materials and methods). To serve as an illustration, we chose the *Delftia* genome in community M04, which had the fewest (186) SNPs among its decoupled strains, which shed light on which functions may be associated with decoupling. The SNPs differentiating these strains were scattered throughout the core genome, with 62% in the coding regions of genes with known functional annotations. Broadly, these genes corresponded to transcriptional regulators such as *gbdR*, transmembrane proteins such as the nitrate transporter *nitT*, and enzymes implicated in central carbon metabolism, such as *aceE*, *murA,* and *ppsA* (*Figure 2e*). The *gbdR* protein is a known transcriptional regulator of amino acid metabolism, glycine betaine catabolism, as well as phosphatase activity in bacteria, and may play a role in differentiating metabolic flux balance between strains (*Wargo et al., 2008*). The *nitT* protein is a member of the major facilitator superfamily (MFS) and a transporter that controls the uptake of nitrate available in the environment (*Maeda and Omata, 1997*). Finally, *ppsA* and *murA* are enzymes that metabolize pyruvate and its derivatives, and may determine how strains catabolize metabolic intermediates before they enter the TCA cycle (*Mikkelsen et al., 2004*; *Marquardt et al., 1992*). Together, this evidence suggests that diversifying in a handful of functional categories is sufficient for strains to decouple in their ecological dynamics.

## Dynamical correlations suggest that interactions are strain-specific

Equipped with the understanding that conspecific strains could display decoupled dynamics, we asked which ecological factors were responsible for the observed differences in strain dynamics. Our experimental setup explicitly controlled for abiotic ecological factors since all communities were grown in the same environmental conditions. Therefore, any dynamical differences between strains must have emerged as a result of biotic factors, such as ecological interactions. To detect and measure interactions between community members, we exploited the fact that all 10 communities were at equilibrium (median ~31% temporal coefficient of variation in the abundance of a species; *Figure 1a*, *Figure 1—figure supplement 1*, *Figure 1—figure supplement 2*). In a controlled environment, interactions between community members at equilibrium are expected to induce temporal correlations between their abundances (*Faust et al., 2015*; *Tikhonov et al., 2015*; *Faust and Raes, 2012*). In contrast, we expect almost no correlations if all abundance fluctuations are purely stochastic (due to neutral drift) (*Hubbell, 2001*; *Descheemaeker and de Buyl, 2020*; *Grilli, 2020*; *Ramsayer et al., 2012*). Motivated by this logic, we measured dynamical correlations between the abundance trajectories of pairs of community members.

To determine which level of taxonomy (strains or species) had the strongest interactions, we compared the strength of correlations between the two levels of taxonomic grouping. Specifically, at the species level, we measured a species–species correlation between each co-occurring species

pair as the magnitude of the correlation between the relative abundances that pair (*Figure 3a*). At the strain level, we only measured correlations between strain pairs belonging to different species, not between conspecifics (which we had already measured in *Figure 2a–c*). To compare strain correlations with their corresponding species correlations, we measured a strain–strain correlation between each species pair; we used the correlation with the highest magnitude among all four possible pairs as the strongest strain pair for a given species pair (*Figure 3a* and Materials and methods).

Remarkably, we found that in the majority of comparisons (76%), at least one pair of strains from two different species was more strongly correlated than the corresponding species themselves (*Figure 3b*, *Figure 3—figure supplement 2*, *Figure 3—figure supplement 3*; p<10$^{-6}$, Wilcoxon signed-rank test adjusted for multiple comparisons). In most cases, even when both strain- and species-level correlations were high, the values of at least one pair of strain correlations were often higher, on average being larger by 0.17 ± 0.01 (one-sample Student's *t*-test, p<10$^{-3}$). To test if this result could arise purely because we compared four pairs of strains for each pair of species, we used a null model in which strains were randomly sampled from the dataset and combined into new mock 'species' (see Materials and methods). This shuffling coalesced unrelated pairs of strains, but preserved the bias in the number of comparisons performed and the abundance data themselves, and revealed that the observed high fraction of strain-dominant interactions was not a statistical artifact (p<10$^{-3}$, permutation test; *Figure 3—figure supplement 1*, *Figure 3—figure supplement 6*). Additionally, interaction networks were, on average, 30% denser at the strain level than the species level (*Figure 3—figure supplement 5* and supplementary text). Finally, in 7% of cases, our analysis revealed so-called 'hidden' interactions, masked at the species level but visible with strains. In these cases, while there was virtually no correlation between species dynamics (*Figure 3c*; Pearson correlation 0.02, p=0.9), the dynamics of their minor strains were strongly correlated (*Figure 3c*; Pearson correlation – 0.71, p<0.01, after a Benjamini–Hochberg correction). These results suggest that coarse-graining communities beyond strains, say into species, decreases dynamical correlations. Strains of the same species often correlate more strongly with strains of another species than with their own conspecifics. One reason for this may be that conspecific strains interact differently with the same community members, making community interactions strain-specific.

## Minimal models support strain-specific, not species-specific, interactions

Two different hypotheses could potentially explain the observation that strain correlations are typically higher than the corresponding species correlations. First, conspecific strains have different interactions, and the increased strain correlations point to systematic differences in, say, the metabolic abilities of conspecific strains. Second, interactions are conserved at the species level (strains are ecologically equivalent, i.e., phenotypically identical), but the increased strain correlations arise from a combination of stochastic effects, such as a higher number of comparisons, measurement noise, and abundance fluctuations within species.

To compare the dynamical patterns expected under both hypotheses, we simulated community dynamics under two scenarios using two minimal consumer-resource models where interactions were mediated by resource competition. In the first model, strains were phenotypically distinct (hypothesis 1; *Figure 3f*), while in the second, strains were phenotypically identical (hypothesis 2; *Figure 3g*). In both models, we simulated the assembly of 10 independent communities under serial propagation in the same environment (see Materials and methods). To model abundance fluctuations, we introduced minor variations in the proportion of resources supplied during each transfer (*Figure 3e*). We encoded competitive interactions through a matrix of resource consumption rates; the difference between the rates of two taxa (hereafter referred to as *D*) inversely controlled the strength of competition between them (*Figure 3d*). In the first model, each conspecific pair had a different set of consumption rates, tuned by the 'competitive distances' *D* between them (*Figure 3f*). In the second model, interactions were instead species-specific, that is, both conspecific strains had identical consumption rates (*D* = 0), with stochastic fluctuations being the only way to alter their frequencies relative to each other (*Figure 3g*).

Remarkably, only one of the models, where conspecific strains have different interactions, produced patterns that were consistent with the data (78% strain correlations greater than species correlations; *Figure 3f*). The alternate model, with phenotypically identical strain interactions, produced

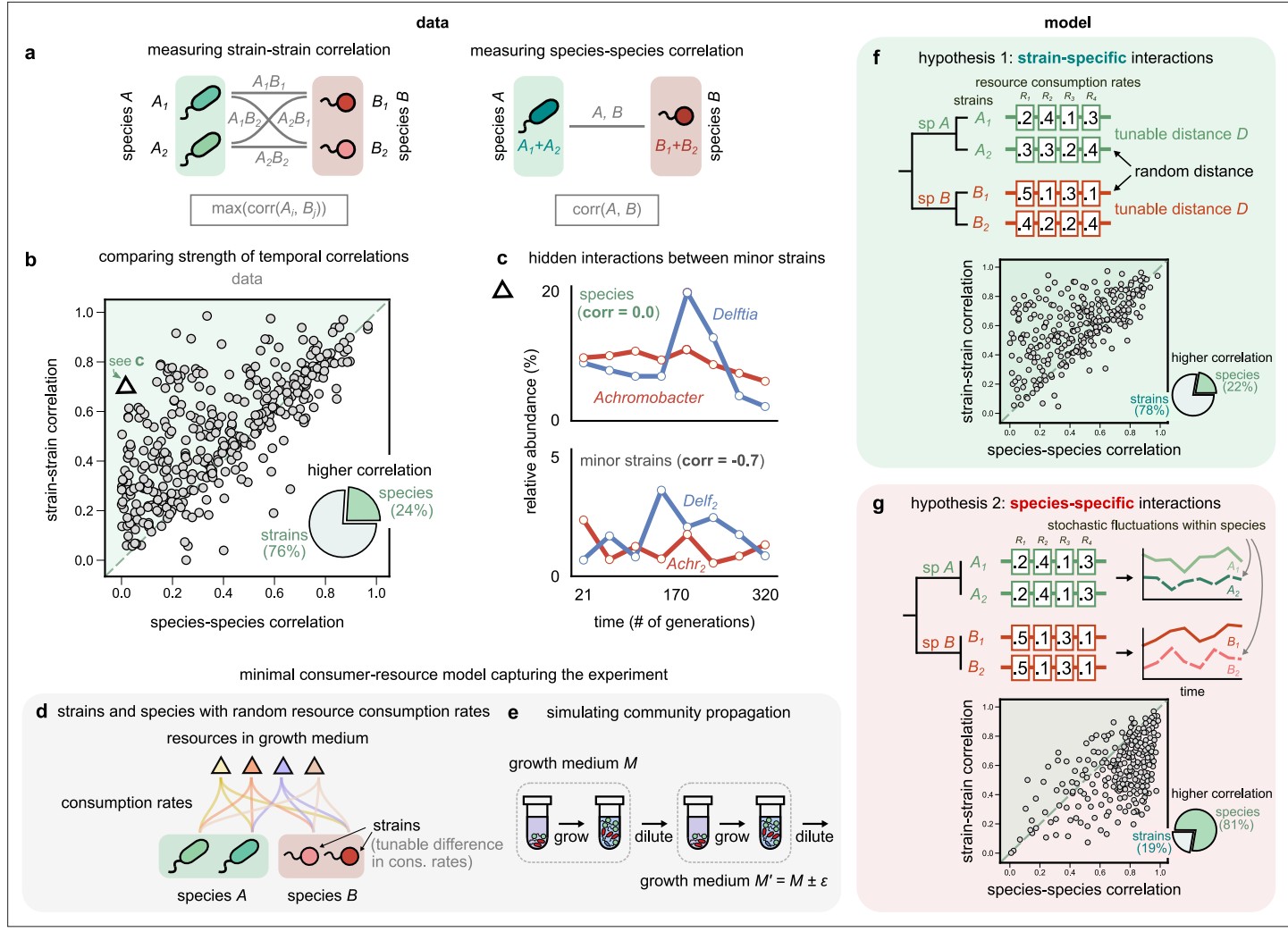

**Figure 3.** Community interactions are strain-specific. (**a**) Schematic showing how we measured dynamical correlations at the strain and species level for a species pair *A* and *B*. For any species pair, we defined strain–strain correlations as the highest magnitude of correlation among all four strain pairs from different species (left), never the same species. The species–species correlation for the same pair was simply the correlation between the two species (right). (**b**) Scatter plot of the dynamical correlation between species in a community and the highest correlation between their corresponding strain pairs. Each point represents one species in one of the 10 communities. The shaded region indicates strain–strain correlations higher than species–species correlations. Inset shows a pie chart of the fraction of points supporting higher strain-level interactions (76%) versus species-level interactions (24%). Triangle indicates a pair of *Achromobacter* (red) and *Delftia* (blue) species, shown in (**c**). Top: relative abundance plots of two uncorrelated species measured over the experiment. Bottom: relative abundances of the minor strains for the same species, which are strongly negatively correlated. (**d, e**) Schematics of our models showing how species are split into strains, with tunable differences in their consumption rates for each resource, as well as the serial dilution protocol that we simulate, where we slightly change the growth medium from transfer to transfer. (**f, g**) Scatter plots of the expected dynamical correlations using our models, (**f**) where strains are ecologically distinct (hypothesis 1) and (**g**) identical (hypothesis 2), similar to (**b**). Schematics of the consumption rate matrices for both models (hypotheses) are also shown.

The online version of this article includes the following figure supplement(s) for figure 3:

**Figure supplement 1.** Null model where we shuffled species–strain associations does not show the observed strain specificity.

**Figure supplement 2.** Dynamical correlations between species and strains do not cluster by community identity.

**Figure supplement 3.** Strain-specific interactions are stronger even when estimating abundances purely from metagenomic reads.

**Figure supplement 4.** Strain-specific interactions are stronger even when using an alternate measure.

**Figure supplement 5.** Interaction networks inferred at the level of species and strains.

**Figure supplement 6.** Examples of shuffled cases where species correlations are higher than strain correlations.

**Figure supplement 7.** Model recapitulates distance-dependent strain decoupling.

**Figure supplement 8.** Geometric interpretation of strain–strain and species–species correlations in our models.

strain correlations that were greater in only 19% of simulated species (*Figure 3g*). Encouragingly, strain–strain correlations produced by the first model could also recapitulate the decoupling between conspecifics beyond a characteristic phenotypic distance (*Figure 3—figure supplement 7*), consistent with our experimental observation of strain decoupling beyond a characteristic genetic distance (*Figure 2c*). We will provide a geometric intuition for this later in the article (Figure 5; *Figure 3—figure supplement 8*). Together, our results show that the patterns observed in the data are consistent with a model in which resource-mediated interactions are stronger between strains, not species.

## Genetic variation in transporters, regulators, and pseudogenes differentiates strains

We finally asked what differentiated strains at the genomic level—in particular, at the level of the core genome, where SNPs can be reliably identified. We had already explored a specific example while studying decoupled strains (*Figure 2e*); we now asked for more general features that differentiated coexisting strains. Most (97%) of the differences between strains in their core genomes were in the form of SNPs; the remaining 3% constituted insertions and deletions (average 4 bp per event; see Materials and methods). We had only one reference genome for each ASV, thus we could not detect changes in the flexible genome, such as gene gains in a specific strain. For all 163 strains, there was no significant bias in the distribution of SNPs between coding and non-coding regions, implying that most (>80%) of the SNPs were found in the coding regions (*Figure 4—figure supplement 1*).

We next asked which functional categories of genes were enriched in strain-differentiating SNPs. For this, we performed a broad categorical analysis, giving us a bird's eye view of the cellular functions likely to be variable in strains (we used annotations and categories from the KEGG database, respectively; see Materials and methods) (*Kanehisa et al., 2017*). Four major functional categories differentiating strains emerged: (1) two-component systems, (2) enzymes involved in carbon metabolism, (3) transcription factors, and (4) transporters (*Figure 4a*). To get a closer look at specific examples of pathways affected by each of these four categories, we examined ASV11, belonging to the genus *Aquitalea* in M04, which contained SNPs in genes from all four categories (*Figure 4b*; *Supplementary file 2*). For two-component systems, we found the *dctD* protein, which, along with *dctB*, regulates the uptake of C4-dicarboxylates such as aspartate, malate, fumarate, and succinate (*Park et al., 2002*). The enzyme *korB* is an oxidoreductase, known to decarboxylate 2-oxoglutarate, a key intermediate in the TCA cycle (*Tersteegen et al., 1997*). The transcription factor *phnF* represses the *phnCDE* operon and regulates the biosynthesis of amino acids such as glutamate (*Aravind and Anantharaman, 2003*). Finally, we found many diverse sugar and amino acid transporters (of the ABC and MFS families), ion transporters like *corA* (*Papp-Wallace and Maguire, 2007*), as well as *iutA*, which mediates siderophore uptake and competition for iron in bacteria (*Figure 4b*; *Krewulak and Vogel, 2008*). Many of these functional categories were specific to the coexisting strains in our communities, and not generic differences between members of the same taxa (*Figure 4—figure supplement 1* and Materials and methods). These examples typify some key functional signatures that differentiate strains and suggest that variation in transporters, regulators, and enzymes in central carbon metabolism may be the basis for strain-specific ecological interactions, for example, by separating the metabolic niches of strains, as in our models.

Another key genomic signature was the presence of several strain-specific pseudogenes in strain genomes. Pseudogenes are genes that contain a premature stop codon and are not expected to produce functional proteins (*Balakirev and Ayala, 2003*). Conspecific strains differed from each other not just by mutations, but also by differential pseudogenization. Not only did one of the two conspecific strains in each pair have several (~5) pseudogenes at the first time point (~23 generations), but we could detect new pseudogenes being generated throughout the experiment (*Figure 4c*, *Figure 4—figure supplement 4*; see Materials and methods for how we detected pseudogenes and assigned them to strains). This suggested that pseudogenization was both extensive and rampant during community evolution. Pseudogenes that differentiated conspecific strains were enriched in two functional categories: (1) motility proteins such as *cheA*, responsible for chemotaxis (*Karatan et al., 2001*), and (2) phage proteins such as *fii* and *gpdD*, known to constitute viral tails (*Figure 4d*; *Temple et al., 1991*). Functions such as chemotaxis and structural viral proteins are not expected to be advantageous for bacterial growth in the environmental conditions of our experiment, and are thus likely to face weak, or no, selection. Indeed, while pseudogenes did not accumulate mutations faster than

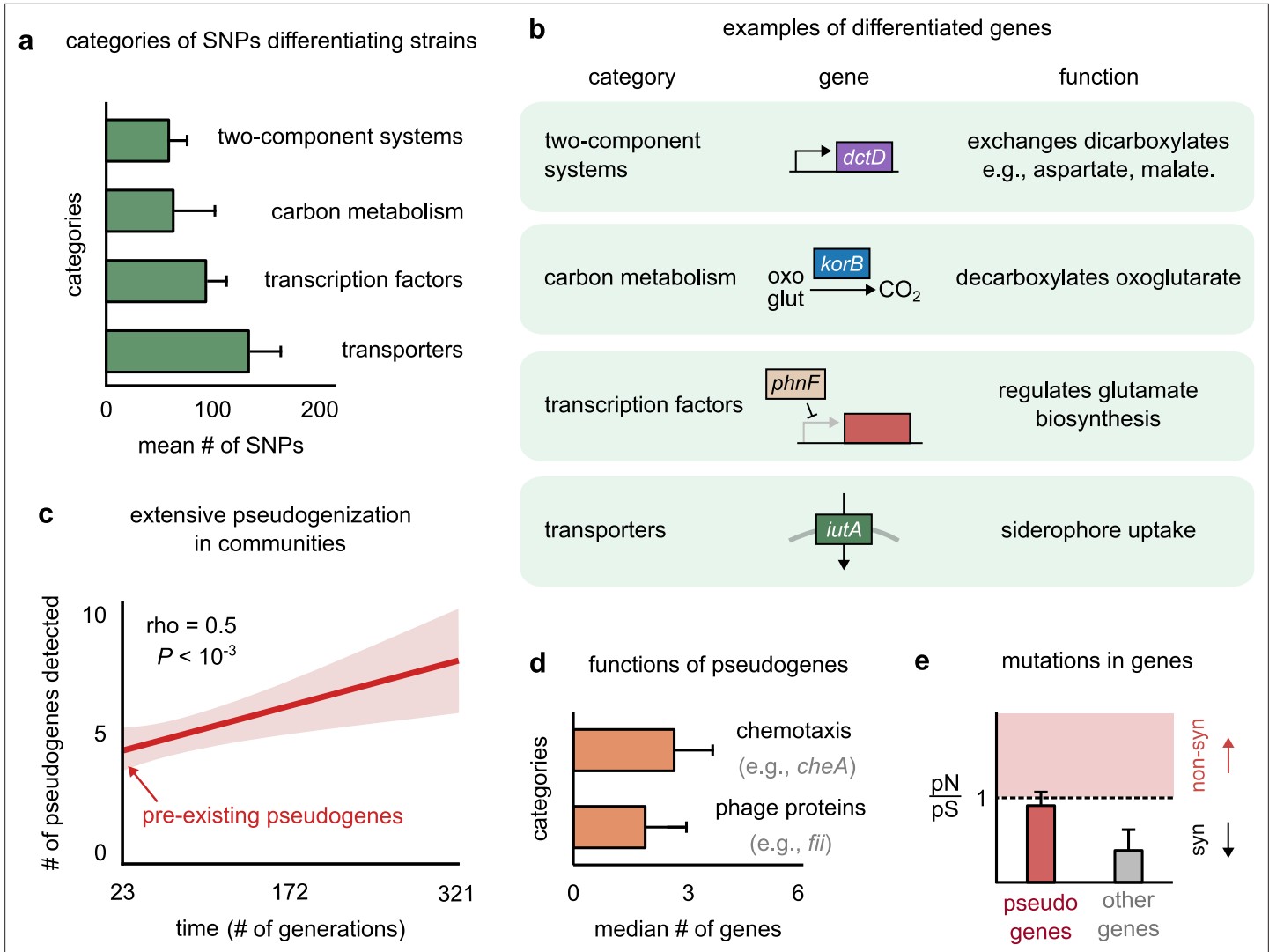

**Figure 4.** Genetic variation in regulators, transporters, and pseudogenes differentiates strains. (**a**) Bar plot showing the four functional categories of genes most enriched in strain-differentiating single-nucleotide polymorphisms (SNPs). The x-axis represents the mean number of SNPs belonging to the category in a strain pair. (**b**) Table showing an example of a gene in each functional category identified in (**a**); the middle column shows a schematic of the gene with its name (italics). (**c**) The average number of pseudogenes detected in strains per strain as a function of time. The solid line shows a linear regression, whose intercept shows the number of pseudogenes detected at the first sequenced time point; the shaded region represents the standard error of the mean (s.e.m.). (**d**) Bar plot showing the two functional categories most enriched in strain-differentiating pseudogenes. The x-axis represents the median number of genes belonging to the category in a strain pair. (**e**) Bar plot showing the mean pN/pS of mutations detected in pseudogenes (red) and all other strain-differentiating genes (gray). Dashed line represents the expected pN/pS under a neutral model. All error bars represent s.e.m.

The online version of this article includes the following figure supplement(s) for figure 4:

**Figure supplement 1.** Most single-nucleotide polymorphisms (SNPs) that differentiate strains are in the coding regions.

**Figure supplement 2.** Mutations accumulate at a similar rate in both pseudogenes and other genes.

**Figure supplement 3.** Functional differences enriched in single-nucleotide polymorphisms (SNPs) differentiating strains of *Aquitalea magnusonii* from the NCBI GenBank database.

**Figure supplement 4.** Dynamics of a de novo loss-of-function (pseudogenizing) mutation.

other genes (*Figure 4—figure supplement 2*), their mutational profile had a pN/pS ~ 1, consistent with neutral evolution (*Figure 4e*). In contrast, other genes showed signatures of purifying selection, with a pN/pS ~ 0.2, significantly lower than 1 (*Figure 4e*). The rampant pseudogenization observed in our communities suggests that evolution tends to deactivate genes that do not contribute to fitness, such as chemotaxis and viral genes in stable, well-mixed environments. The end products of such

nonfunctional pseudogenization and functional genomic variation may lead to further phenotypic differences between strains, allowing even highly related strains to show decoupled dynamics.

## Discussion

Understanding the role of strains in microbiomes is crucial to the study of microbial ecology and evolution (*Van Rossum et al., 2020*). Here, we addressed this question by propagating microbial communities from pitcher plants for more than 300 generations. We found that strains that were preexisting genetic variants belonging to the same species were the key determinants of long-term community dynamics. Differing by between 1 and ~10,000 SNPs, strains imparted both compositional and dynamical variability to communities. Compositionally, we found that communities were most variable in terms of strains, with each community carrying a unique set of strains even when they carried the same species. This variability was likely the result of two chance events: stochastic sampling and laboratory selection. Dynamically, once the communities settled into unique equilibria, we found that the dynamics of even extremely closely related strains, with as few as 100 SNPs, could be decoupled from each other. Even when a species' abundance appeared relatively stable over time, strains comprising it could be highly dynamic. Such strains were temporally correlated with other strains belonging to different species. Mathematical consumer-resource models capturing our experiment suggested that these observations and patterns were consistent with resource-mediated competitive interactions between strains, not species.

Our observation that correlated fluctuations among existing strains dominate evolutionary dynamics is perhaps somewhat different from one's naïve expectation, anticipating new mutations and recombination events to be the dominant forces driving evolution within communities (*Plucain et al., 2014*). Once our communities reached a stable equilibrium, their evolution was marked by the dynamics of the preexisting strains, which could stably coexist for hundreds of generations. Thus, in both cases, either by mutation and recombination (in single-species laboratory evolution) or by assembly (in complex communities as in this article), the dynamics of populations are determined by the ecological interactions between fine-scale genetic variants, or strains (*Good et al., 2017*; *Roodgar et al., 2009*). Moreover, in both cases, the interactions between strains are likely resource-mediated since the genetic differences between ecologically divergent strains are concentrated in metabolic genes (*Plucain et al., 2014*). Taken together, these results suggest a striking parallel between the eco-evolutionary dynamics of both isogenic populations as well as complex communities.

The patterns we observed, such as conspecific strain decoupling and higher strain correlations beyond species boundaries, underscore the emerging view that strains are the most dynamic and interactive units of microbiomes (*Roodgar et al., 2009*; *Leventhal et al., 2018*; *Goyal, 2018*). Our study has one major advantage over previous work where natural communities, such as human gut microbiomes, were sampled over time without environmental control. That is, that, by experimentally propagating natural samples, we shielded the communities from external host-induced perturbations such as the migration of new genetic variants or immune system control. Therefore, any observed community dynamics were a result of intrinsic causes, such as strain-specific interactions, not extrinsic causes, such as host-induced shifts. The reproducibility of these patterns across 10 independent replicate communities further strengthens our findings.

Finally, our results raise a question about the appropriate level at which to monitor microbial community composition. On one hand, our results show that strains as few as 100 SNPs apart can have independent ecological dynamics and interactions, and as such, represent distinct ecological variables. On the other hand, consistent with previous work, our results also show that the presence of strains is highly variable across communities, even in the same abiotic environmental conditions (*Leventhal et al., 2018*). Therefore, coarse-grained assemblages (e.g., members of the same taxonomic family) should better represent the relationship between metabolic niches and community composition, especially for short-term community assembly dynamics (*Goldford et al., 2018*; *Louca et al., 2016*). But it is not clear how such assemblages should be defined in general, and what type of functional redundancy they capture. This is because there are numerous definitions of what constitutes 'function.' In the study that preceded this article, we found that the substrate consumption profile of the community was strongly correlated with community composition (at the 16S rRNA level), indicating thus that there is, at best, weak functional redundancy in the community if substrate preference is what we define as 'function' (*Bittleston et al., 2020*). However, if what we care about is

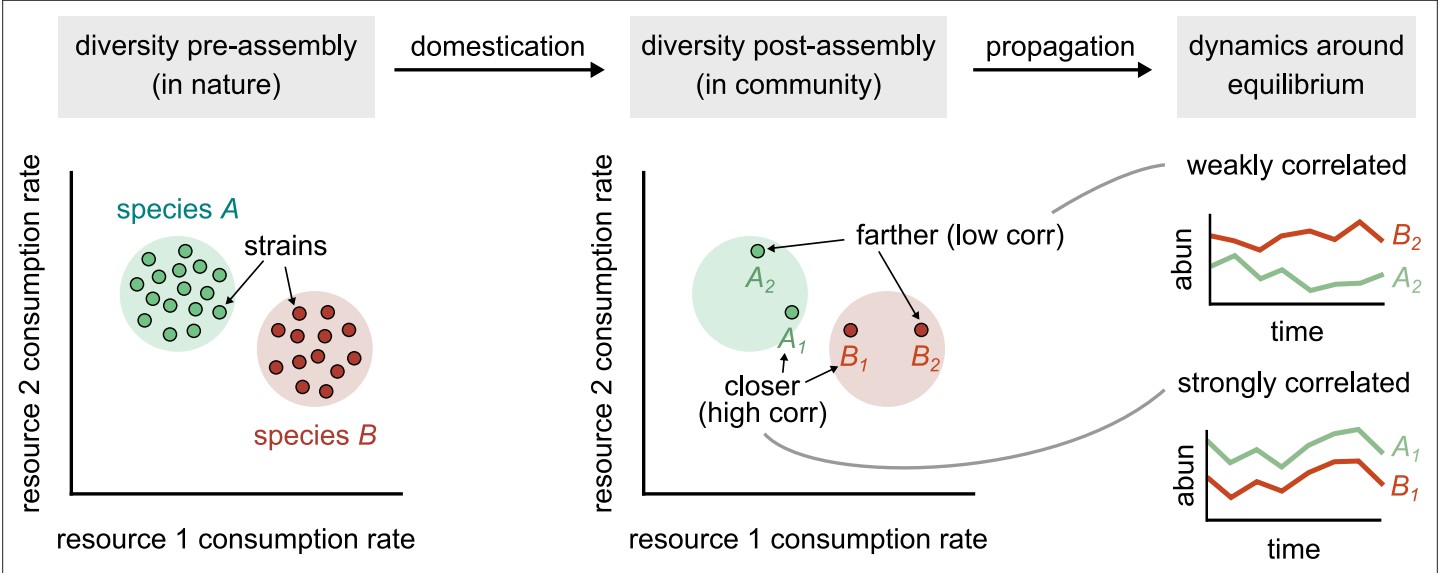

**Figure 5.** Conceptual model of strain-dominated long-term dynamics. (Diversity pre-assembly) Schematic showing the phenotypes (e.g., resource consumption rates) of strains belonging to two different species, *A* (green) and *B* (red), in a large strain pool in nature. Each point represents the consumption rate of a particular strain on two hypothetical resources. (Diversity post-assembly) When assembled in a community (e.g., domesticated in the lab), a subset of the strains from each species (here, two from each species) may survive and coexist once the community reaches equilibrium. (Dynamics around equilibrium) The strains from each species influence each other's long-term dynamics around equilibrium. Strains from different species that are closer in phenotype space ($A_1$ and $B_1$) will display strongly correlated dynamics while phenotypically distant strains ($A_2$ and $B_2$) will be weakly correlated.

the conversion of organic carbon to $CO_2$, for instance, community composition is indeed highly redundant (most species in the community could perform aerobic respiration and yield similar amounts of $CO_2$). More work is needed in this arena to clarify how relevant ecological functions diverge across the microbial phylogeny. However, at the very least, we can say that if our questions pertain long-term dynamics *Figure 5*, or (eco)-evolution, strains are the appropriate variables with which to study the system.

## Materials and methods
### Experimental methods
#### Sampling, experimental design, and DNA extraction
We collected aquatic samples from 10 healthy *S. purpurea* pitchers at Harvard Pond (Petersham, MA), filtered them through 3 μm syringe filters, combined them in a 1:1 ratio with sterilized cricket media, and grew them in 48-well plates in a 25°C incubator. The sampling and experimental design was identical to, and is described in detail in, *Bittleston et al., 2020*. For the first 63 days (21 transfers), every 3 days each sample was mixed well and 500 μL was transferred to a new plate with 500 μL of sterile cricket media. After this point, we shifted to sampling every 7 days using a larger dilution ratio of 1:100, with 20 μL of each community mixed into 1980 μL of cricket media. At every transfer, we froze a portion of each sample at –80°C for later DNA extraction and sequencing. We extracted and quantified DNA as described in *Bittleston et al., 2020* using the Agencourt DNAdvance kit (Beckman Coulter) the Quant-iT PicoGreen dsDNA Assay kit (Invitrogen), respectively, from transfers 23–67. The intermediate transfer between the two serial propagation regimes, numbered 22, was not sequenced.

#### Strain isolation
Individual strains were isolated from 5 of the 10 microcosms (M03, M05, M07, M09, and M10) by plating the culture fluid and picking around 100 colonies per microcosm. Details of the isolation methods and preliminary strain identification are described in *Bittleston et al., 2020*. We chose a set

of 33 diverse strains that were well-represented in the amplicon sequencing data from the first 63 days and extracted DNA using the same procedure as the community samples (*Supplementary file 1*).

## Sequencing

Amplicons were sequenced at the Environmental Sample Preparation and Sequencing Facility at Argonne National Laboratory on a MiSeq targeting the V4 region of 16S rRNA using the 515F and 806R primers (*Caporaso et al., 2012*; *Caporaso et al., 2011*) and using the same procedure as described in *Bittleston et al., 2020*. Metagenomes from all 10 communities were sequenced across eight evenly spaced time points, resulting in 80 samples. Genomic and metagenomic libraries were prepared at the BioMicro Center at MIT using NexteraFlex and sequenced on a NovaSeq SP500 run, aiming for 20× more sequencing depth for strains relative to metagenomes.

## Bioinformatic methods

### Species abundance estimation

We defined each species in our analysis as a unique amplicon sequence variant (ASV). Our definition of species as exact 16S rRNA variants, or ASVs, is consistent with recent community analyses, which increasingly prefer analyzing communities at the level of ASVs over the historical operational taxonomic units (OTUs). To analyze the amplicon sequencing data, we used the same ASV assignments and sequence analysis procedure described in *Bittleston et al., 2020*. Each species (or ASV) was identified with a unique ~250-bp-long sequence. In each sample, we estimated the relative abundance of a species as the number of reads that mapped to it, normalized by the total number of reads in the sample. We estimated the number of elapsed generations after each cycle in the experiment as $\log_2(D)$, with the dilution factor $D$ = 100. Unless stated otherwise, we performed all subsequent analyses using Python v3.7.3 and the NumPy and SciPy packages (*Harris et al., 2020*; *Virtanen et al., 2020*).

### Reference genome database construction

We built a reference genome database by sequencing 33 out of the 100 isolates extracted from our communities (described above). After trimming raw sequencing reads with Trimmomatic 0.36 (*Bolger et al., 2014*), we assembled genomes using SPAdes v3.13 (*Deng et al., 2013*). We removed all assembled contigs less than 200 bp in length. We then used the Genome Taxonomy Database (GTDB-Tk) to assign genus-level identity to each genome and generate a phylogenetic tree (*Chaumeil et al., 2019*).

### Read mapping

After trimming reads with Trimmomatic 0.36, we mapped each read against our reference genome database using Minimap2 v2.17 (*Li, 2018*). Since our reference genomes were assembled using isolates from the same communities as the reads, we used stringent settings for short reads (**-ax sr**) and to ensure unambiguous read mapping only kept mapped reads longer than 100 bp with the highest MAPQ quality score of 60. Further, we filtered out any remaining reads that mapped equally well to more than one location in the reference database. We mapped reads sample by sample, each sample corresponding to a unique time point from among one of our 10 communities.

To minimize read stealing, we took three crucial steps. First, we only used reads that mapped to the 33 reference genomes we sequenced and assembled from the communities in this study, ensuring that they were representative of the species being studied. Second, we filtered out all reads mapping to genes that were >98% identical across any reference genomes in our database. Finally, we filtered out those reads that mapped to genes with anomalous coverage (either less than half the median coverage in a sample or greater than thrice the median coverage in any sample). The resulting fraction of reads mapped to each species, after normalizing for genome length, was well-correlated with its relative abundance estimated independently using 16S rRNA sequencing, suggesting that read stealing was minimal (mean correlation 0.8; *Figure 1—figure supplement 10*). Further, using species abundances estimated from metagenomic sequencing, rather than 16S sequencing, did not affect our main results (*Figure 3—figure supplement 3* shows a version of *Figure 3b* using only metagenomic abundance estimates).

## Variant calling

To identify genetic variants using the aligned reads, we used the Bayesian genetic variant detector FreeBayes v1.3.2, which prioritizes variation in literal read sequences over variation in their exact alignment (*Garrison and Marth, 2012*). To calibrate the variant caller for haploid prokaryotic genomes, we used the settings **--ploidy 1 --haplotype-length 0 --min-alternate-count 1 --min-alternate-fraction 0 --pooled-continuous --report-monomorphic -m 60 -e 1,000**. For downstream analysis, we only considered biallelic (only one alternate allele observed at the site, >97%) SNPs that had a Phred quality score ≥20 and a local read depth ≥10 (i.e., allele frequency least count: 10%). To avoid sequencing errors and read mapping artifacts, once an SNP was detected at a certain time point in a community, we required that it be detected across all subsequent time points, or until the time point at which the species containing it reached a relative abundance of zero.

## Strain identification

For each community, we chose all SNPs detected at the first time point and partitioned them among the 33 reference genomes (unique 16S rRNA sequences, or species) they were detected in. For each set of SNPs detected in a species, we clustered the frequency trajectories of their major allele (reference or alternate allele, whichever was higher) using dynamic time warping, a generalized *k*-means clustering algorithm for time-series data (*Deng et al., 2020*). We performed clustering for one, two, and three clusters for each species. In all cases, a third cluster had a low Euclidean distance (mean < 0.15) from and was visually indistinguishable from one of the other two clusters (*Figure 1—figure supplement 5*). We thus rejected it. A second cluster was either also visually indistinguishable from the previous clusters, or comprised a set of SNPs which with nearly fixed allele frequencies (>0.9). We inferred the latter clusters as belonging to those loci that were shared by all conspecific strains in the sample but differentiated them from the reference genome. If the latter type of cluster was present, we assigned two clusters to each species. If not, we assigned one cluster. The first two clusters always accounted for >80% of SNPs in a species, which increased our confidence in cluster assignment. Clustering using an alternate method, such as the unweighted pair group method with arithmetic mean (UPGMA) clustering on the data (using the Euclidean distance between SNP trajectory correlations as a distance metric), yielded similar results (*Figure 1—figure supplement 6*). Always, one cluster consisted of SNPs differentiating the two strains, and the second cluster consisted of shared SNPs between them (*Figure 1—figure supplement 7d*). In a few cases, there was a set of unclustered, loose SNPs (e.g., *Figure 1—figure supplement 6b*), which we verified also corresponded to nearly fixed SNPs (allele frequency ~1 at all times; *Figure 1—figure supplement 5b*), which, using the pigeonhole principle, we interpreted to be mutations common to both strains. The coverage-dependent noise in SNP frequency measurements creates a relatively wide cluster, which we suspect somewhat affects our ability to discern several strains. However, a similar small number of strains (usually two) are commonly found in other communities via metagenomic techniques.

## Strain abundance estimation

We identified each strain as being represented by its reference genome along with the cumulative set of SNPs in both clusters. To assign each strain as major or minor, we estimated their frequency within each species. For this, we only used the allele frequencies of SNPs in the first cluster. Specifically, we calculated the average allele frequency weighted by the local read depth for each SNP in the cluster. The strain with a higher weighted mean frequency at the first time point was classified as the major strain. The abundance of each strain was calculated as its frequency at the time point (between 0% and 100%) times the relative abundance of the species in the community at that time point. Thus, each strain pair partitioned the abundance of its species into two abundance trajectories, that of a major and a minor strain.

## Functional annotation of strain-differentiating SNPs

We performed a categorical enrichment analysis of strain-differentiating SNPs (first SNP cluster, described above). First, we extracted those SNPs detected in the genes of each reference genome, on average 82% of SNPs. We annotated the proteins corresponding to each gene using eggNOG-mapper v2 (*Huerta-Cepas et al., 2017*), with the parameters **--go_evidence non-electronic --target_orthologs all --seed_ortholog_evalue 0.001 --seed_ortholog_score 60**. After filtering out

genes with unknown functional annotations such as hypothetical proteins and proteins with domains of unknown functions, as well as genes with no known KEGG Orthology (KO) group (30% of SNP-containing genes), we asked which functional categories of genes had more SNPs than expected by chance. Each KO was associated with a specific functional category in the associated BRITE hierarchy. For the BRITE category 'Enzymes,' which is extremely broad, we manually chose finer functional categories, such as carbon, nitrogen, and sulfur metabolism, based on what compounds the enzymes acted on. For each functional category containing at least one SNP, we calculated an enrichment score, given by the inverse of the p-value of the observed number of SNPs in the category when compared against the expected number of SNPs (given by a binomial distribution with success rate equal to the fraction of the genome corresponding to that category of genes; we accounted for the number and length of genes in each category while computing the expected distribution). Gene annotations and names in examples were derived manually using the KEGG and UniProt entries corresponding to each gene's KO numbers.

As a control, we measured the functional differences among conspecific strains from the NCBI GenBank database, which did not coexist in the same community. The results of such an analysis could depend strongly on which species we chose. To ensure a fair comparison, we chose a species from the most common genus in our communities, *Aquitalea magnusonii*. There were five strains whose genomes were available, all of which we used for our analysis. We arbitrarily chose one of the strains (GCA_003202035.1) as the reference and repeated our analysis exactly as described for the strains in our communities.

## Pseudogene detection and analysis

To detect pseudogenes, we used those strain-differentiating SNPs that were localized in genes (described above). Specifically, we identified the codon-level changes engendered by each SNP, and a pseudogene was identified as one that resulted in a premature stop codon. Like all SNPs, we required that this stop codon-enabling SNP be detected at all subsequent time points once detected. For functional analysis, we used the same procedure as described above but restricted to pseudogenes. To detect mutations in genes, we counted those SNPs that were not detected at the first metagenomic time point but detected at later time points; like other SNPs, once detected, we required mutations to be repeatedly detected at all subsequent time points (see *Figure 4—figure supplement 4* for an example). If an SNP was not detected at the first time point, but was detected later at a frequency consistent with one (or both) of the strains within a species, then we could not confidently assign it as a de novo or preexisting mutation, and we thus refrained from assigning it to either category. To assign a pseudogene to a specific strain, we checked whether the allele frequency of the loss-of-function SNP was consistent with the frequency of one of the two strains. When it was consistent with only one of the strains, but inconsistent with the other, we assigned it to the consistent strain. If the SNP frequency was consistent with both strains (or inconsistent with both strains), then we did not assign it to a unique strain. To calculate pN/pS for each gene (at the final time point), we accumulated all SNPs detected in that gene and classified them as synonymous or nonsynonymous based on whether they led to an expected amino acid change; we then calculated pN/pS using standard techniques.

## Statistical methods and models

### Community compositional variability

We measured the variability in community composition across our 10 communities at different taxonomic levels. At each taxonomic level, we partitioned all community members into groups; members within a group differed at that taxonomic level but shared a common ancestor just one taxonomic level above. As an example, to measure variability at the species level, we partitioned all members into groups; each group contained different species belonging to the same genus (say all three species belonging to the genus *Aquitalea*). After partitioning, we measured the probability with (or frequency at) which two members of a group co-occurred in a sample (how often two different *Aquitalea* species co-occurred). This gave us a measure of how different two communities were at a given taxonomic level (in this example, to what extent different communities had different species of the same genus).

To normalize the observed probability of co-occurrence against the expected probability at each level, we repeated the calculation by randomly shuffling member labels across groups. This procedure

destroyed any phylogenetic relationship between members of a group, but preserved both the number of groups and their sizes.

## Divergence time between strains

From previous literature, we estimated the mutation rates of soil bacteria ($10^{-8}$ per nucleotide per generation), genome size of roughly $10^6$ bp, and an average doubling estimate of ~100 generations per year for soil bacteria in the wild (*Ochman et al., 1999*). Using these estimates and ideas from molecular clock analyses, which suggest that the divergence rate of neutral mutations in a population is equal to the per individual mutation rate (independent of population size), we calculated the average fixation time for one mutation to be about 1 year. This estimate is consistent with known mutation accumulation rates in bacterial genomes (*Gibson and Eyre-Walker, 2019*). The pitchers of *S. purpurea* that we sampled were a few months old (between 3 and 4 months; we sampled in September and new pitchers open in June). Based on these rough estimates, we believe that it is highly unlikely that any pair of strains, which we observed to coexist, diverged during the lifetime of a single pitcher.

## Strain–strain coupling

We measured a strain–strain coupling between each conspecific pair of strains (belonging to the same species) that co-occurred in a community. For each conspecific strain pair, we calculated their temporal abundance trajectories, that is, their relative abundances at all eight time points (described above), and then measured the magnitude (or absolute value) of the Pearson correlation coefficient between them. We measured only the magnitude of each correlation (regardless of its o-value) because we were interested only in the extent of covariance between two conspecific strains, and not in the statistical significance of their association. Further, we used the absolute value of each correlation and ignored their sign for simplicity since there were very few negative correlations, most of which had a low magnitude (<0.3; *Figure 2—figure supplement 3*). The latter were correctly classified as decoupled even under the simpler magnitude-only definition of coupling. We verified that using a nonparametric covariance measure such as the Spearman correlation did not affect our results (*Figure 2—figure supplement 2*).

Since we metagenomically sequenced the communities roughly once every 37 generations (298 generations evenly sampled eight times), we could not resolve dynamics finer than that timescale. As a result, while our method would be able to resolve a time lag between strain dynamics, the time lag could be no finer than our limit of resolution, that is, 37 generations. No strain dynamics lagged or led the other by greater than 37 generations. Since the presence of any undetected correlations between strains at finer temporal resolutions would only strengthen our claims, we believe that our results are robust to our limited temporal sampling. Further, while our correlation-based method would be able to detect higher-order interactions between strains, we would require data from many more communities than the 10 we sampled here. This is because resolving higher-order interactions between any trio of strains requires tracking the dynamics of each pair of the trio, both in the presence and absence of the third strain.

## Interactions between strains and species

We measured putative signatures of interaction in each community separately at the level of species and strains. At each level (say species), we first calculated the temporal abundance trajectories of all members, that is, their relative abundances at all eight time points (described above). Then, for each inter-species pair, we then measured the Pearson correlation coefficient between their abundance trajectories (similar results with Spearman correlations, *Figure 3—figure supplement 4*). We used the magnitude of the correlation as a proxy for interaction strength. Relative abundance data can exhibit spurious correlations because of finite sampling and the constraint that abundances must sum up to 1. To account for this while detecting the presence of an interaction, we calculated the statistical significance of each correlation against an expected correlation distribution, described as follows.

## Null model to detect interactions

To estimate the expected correlation distribution between the abundance trajectories of two species (or strains) in a community, we used a null model to simulate community abundances. The null model

made two key assumptions: the abundance fluctuations of each member were independent of each other (as expected when there are no interactions) and followed a gamma distribution, which empirically describe abundance fluctuations in microbial communities (*Grilli, 2020*; *Ramsayer et al., 2012*). For each species , we used the observed mean ($\bar{x}_i$) and variance ($\sigma_i^2$) of its abundance, $x_i$, from all our communities to fix the parameters of its expected abundance distribution $\mathcal{P}_i(x)$ as follows:

$$\mathcal{P}_i(x) = \frac{1}{\Gamma(\beta_i)} \left( \frac{\beta_i}{\bar{x}_i} \right)^{\beta_i} x^{\beta_i - 1} \exp\left( -\beta_i \frac{x}{\bar{x}_i} \right)$$

Here, $\beta_i = \bar{x}_i^2 / \sigma_i^2$. We then simulated the abundance trajectories of each community with the same species or strains as laboratory communities. At each time point, we randomly picked the abundance of each community member from its expected abundance distribution. After picking the abundance of all members, we renormalized them (divided each abundance by the total sum) to obtain relative abundances. We then measured the correlations between each pair of species (or strains) using this synthetic abundance data and using the same procedure described for real data. We repeated the simulations and correlation measurements 1000 times to build an expected correlation distribution. To measure the statistical significance of a specific correlation observed between two species or strains, we calculated its p-value using this expected distribution, that is, the probability of obtaining this correlation by chance. We called an interaction 'present' if its p-value was <0.05 (*Figure 3—figure supplement 5*). Since the expected distribution already accounted for multiple comparisons, we did not perform an additional correction.

## Null model to control for phylogenetic association

To control for the association between species and strains while comparing species–species and strain–strain interaction strengths, we calculated the fraction of cases where strains were more interactive than species in a null model. To do this, we shuffled the association between species and strains by redistributing strains across species. In each community, we randomly shuffled strain labels, thus splitting conspecific strains across different species and coalescing interspecific strains into the same species. We repeated our measurement of strain–strain and species–species interaction strengths on these shuffled data and calculated the fraction $\mathcal{F}$ of cases where the magnitude of correlation between a pair of strains from different (relabeled) species was higher than the correlation between the (relabeled) species themselves. We repeated this shuffling (permutation) 1000 times, thus obtaining a null distribution $\mathcal{P}(\mathcal{F})$, with a mean 64% and standard deviation 3%, and a corresponding p-value for our experimentally measured fraction $\mathcal{F}_{obs}$ = 76%, that is, the probability of observing a fraction $\mathcal{F}$ equal to or greater than 76% (*Figure 3—figure supplement 1*).

## Measuring eco-evolutionary influence from strains on species

Communities can be characterized by two kinds of dynamics: (1) ecological dynamics describe the changes in community composition at the level of species while (2) evolutionary dynamics describe the changes in the composition of each species at the level of the genotypes or strains that constitute them. A hallmark of microbial communities is that ecological and evolutionary dynamics are often coupled to each other due to the short generation times of microbes. Namely, changes in strain frequencies affect subsequent changes in species' relative abundances, which then drive further changes in strain frequencies. This phenomenon has been extensively studied in two-species communities and has been termed eco-evolutionary feedback. Inspired by this idea, we wondered to what extent the changes in species' relative abundances were driven by the changes in the frequencies of their underlying strains. Our communities, which had much higher species diversity (~10 species per community) and 10 independent replicates, allowed us to perform a more extensive test of this idea. Due to the low temporal resolution of our measurements (eight time points per community), we would not be able to discern a continuous feedback between the species and the underlying strain dynamics, but would still be able to detect some signature of one influencing the other. For example, in *Figure 2d*, the dynamics of an *Aquitalea* species could be almost entirely explained by the growth of its minor strain. We termed such a phenomenon—where the abundance fluctuations of a species coincided with the fluctuations in its underlying strain frequencies—'eco-evolutionary influence.

To quantify how common this phenomenon was in diverse microbial communities, we calculated the eco-evolutionary influence for all species across all 10 communities. Specifically, we calculated the correlation between two quantities: (1) the relative abundance of each species trajectory over time and (2) the frequency of its major strain or genotype of that species over time. A high correlation would indicate the presence of an eco-evolutionary influence (*Figure 2d*) while a low correlation would not. To ascertain the boundary between a low and high correlation, we employed a null model. Namely, in this null model, we shuffled all species and major strain trajectories, and measured their correlations to generate an expected distribution of eco-evolutionary influence (*Figure 1—figure supplement 9*, gray).

We found that a majority of species trajectories (51%) had a greater eco-evolutionary influence than expected by chance (*Figure 1—figure supplement 9*, red), suggesting that eco-evolutionary influence is common in complex communities.

## Constructing community interaction networks

We looked for the presence of interactions between members for each community separately. To do this, we inferred two interaction networks: one at the level of species and the other at the level of strains. To infer an interaction network, say at the species level, we measured all pairwise correlations between the abundances of species in the community. Two species were said to interact if the correlation between them was statistically significant compared with an ecological neutral model. Briefly, this model computed the expected distribution of correlations between noninteracting members of a community by simulating their abundance trajectories under known empirical laws using only the mean and variance of their observed relative abundances (see Materials and methods). Importantly, even at the level of strains, we specifically looked for interactions between strains belonging to different species, not between strains of the same species.

To illustrate our results, we focused on community M07 as an example (*Figure 3—figure supplement 5*). We found that the strain-level interaction network of this community was 90% denser in interactions (38 interactions across 24 strains; density measured as the number of interactions per node) than the species-level network (10 interactions across 12 species). In fact, this observation of denser strain-level interaction networks was true for all communities (mean ~ 70% denser networks at the strain-level). Most strain-level networks revealed interactions that could only be observed at the level of strains, not species (e.g., *Figure 3—figure supplement 5*, *Rh*). These results suggested that in terms of the presence/absence of interactions community interactions were likely strain specific.

Note that in this analysis we were interested in measuring the density of interactions as the number of interactions (edges) detected per node (strain or species), not the number of interactions (edges) per possible edge, which would control for a fixed false-positive rate of observing interactions. This is because we were interested in measuring how many other community members each constituent strain or species interacted with. Specifically, with this analysis, we wanted to ask if strains had more interacting partners than species, on average.

## Measuring functional differences between generic strains

As a control, we measured the functional differences among conspecific strains from the NCBI GenBank database, which did not coexist in the same community. The results of such an analysis could depend strongly on which species we chose. To ensure a fair comparison, we chose a species from the most common genus in our communities, *A. magnusonii*. We found that the functions differentiating *A. magnusonii* strains had very little overlap (~10%) with those differentiating coexisting *Aquitalea* strains in our communities. Enriched functions in the controls comprised translation factors, motility genes, antimicrobial resistance genes, and transporters (*Figure 4—figure supplement 3*). Even though transporters were variable across both kinds of strains, upon closer inspection, we found differences in the types of transporters enriched in our strains and controls. Unlike coexisting strains, where most variable transporters belonged to sugar, amino acid, and siderophore transporters, among non-coexisting strains, variable transporters instead belonged to outer membrane porins like *bamA*, metal ion transporters like *cbiN*, and multidrug resistance pumps like *emrE* and *qac*.

## Minimal consumer-resource models of strain interactions

To compare the dynamical patterns expected under both hypotheses, we simulated community dynamics under two model scenarios using two similar consumer-resource models where interactions were mediated by resource competition.

We initially populated each community with a randomly generated ensemble of species, consisting of 50 species or 100 strains. For simplicity, each species was composed of two strains. We encoded competitive interactions through a matrix of resource consumption rates; the difference between the rates of two taxa inversely controlled the strength of competition between them. In the matrix, each species was represented by a randomly chosen vector of consumption rates. Each set of consumption rates was represented by a vector of dimension 30 (equal to the number of resources), sampled uniformly from the unit simplex, such that all resource consumption rates summed to 1. This well-studied assumption is common to consumer-resource models since it mimics a fixed 'enzyme budget' for each strain and allows strains to coexist (*Goldford et al., 2018*; *Posfai et al., 2017*; *Goyal and Maslov, 2018*). This assumption greatly simplified the analysis of our model since we were primarily interested in studying the dynamical correlations between strains rather than their coexistence. We simulated the model in discrete growth-dilution cycles, where in each cycle we simulated the dynamics of strains and resources using the standard consumer-resource equations with resource competition as follows:

$$\frac{dN_\alpha}{dt} = \sum_i C_{\alpha i} N_\alpha R_i,$$

and

$$\frac{dR_i}{dt} = -\sum_\alpha C_{\alpha i} N_\alpha R_i,$$

where strains' population sizes are represented by $N_\alpha$, resource concentrations by $R_i$, and $C_{\alpha i}$ represents the consumption rate of strain α for resource *i*.

Mimicking the experimental protocol, we simulated the assembly of 10 independent communities under serial propagation in the same balanced resource environment, with a dilution factor of 100, until all communities reached an equilibrium. To obtain abundance fluctuations around equilibrium, we utilized the fact that the experimental medium was complex, not well-defined. While the medium had an average composition, the amount of each resource in it would have differed slightly from transfer to transfer. To model this, we introduced minor variations in the relative proportion of resources supplied during each transfer, such that all communities experienced the same resource environment on average. Specifically, we chose a medium with 30 resources, such that all resources were supplied in equal proportions on average, with the total resource supply summing to 1 in units of microbial biomass. Variations in the resource supply were modeled as Gaussian-distributed noise with mean 0 and variance 0.05. We verified that changing the number of resources did not quantitatively affect our results.

When interactions were strain-specific, each conspecific pair had a different set of consumption rates, with a chosen 'competitive distance' *D* between them, where *D* represents the average difference in the consumption rates of any one resource. Here, each strain was represented by a set of consumption rates that summed to 1, but such that its Euclidean distance from its conspecific strain's consumption rates was *D*. Similar to our observation that strains had a broad distribution of genetic distances, pairs of strains in our model had a wide range of competitive distances, *D*. Specifically, we chose 15% of strain pairs with a distance *D* = 0.01, 20% with *D* = 0.1, 50% with *D* = 0.3, and 15% with *D* = 1.0. This choice was arbitrary but somewhat mimicked the observed distribution of genetic distances between strains.

When interactions were instead species-specific, both conspecific strains had identical consumption rates (*D* = 0), with stochastic noise being the only way to alter their abundances. To implement noise, we simulated a random walk of the relative frequency of each strain pair within a species at each generation, with mean zero and variance $\sqrt{\left(\frac{v}{N}\right)}$, where *N* represents the relative abundance of the species to which the strains belonged, and v̂ is the fit from the data, as the variance in the relative frequency fluctuations of strains, averaged across all species and communities.

## Acknowledgements

We thank Michelle Oraa Ali for their illustration in *Figure 1*. We are grateful to Shaul Pollak for stimulating discussions about the models. This work was supported by the Gordon and Betty Moore Foundation Physics of Living Systems Fellowship grant # GBMF4513 (AG), the Human Frontiers Science Program grant # LT000643/2016L (GEL), the James S McDonnell Foundation Postdoctoral Fellowship award # 220020477 (LSB), the NSF-DEB grant # 1655983 (OXC), and the Simons Foundation Collaboration: Principles of Microbial Ecosystems (PriME) award # 542395 (OXC).

## Additional information

### Funding

| Funder | Grant reference number | Author |
|---|---|---|
| Gordon and Betty Moore Foundation | GBMF4513 | Akshit Goyal |
| Human Frontiers Science Program | LT000643/2016-L | Gabriel E Leventhal |
| James S. McDonnell Foundation | 220020477 | Leonora S Bittleston |
| National Science Foundation | 1655983 | Otto X Cordero |
| Simons Foundation | 542395 | Otto X Cordero |

The funders had no role in study design, data collection and interpretation, or the decision to submit the work for publication.

### Author contributions

Akshit Goyal, Conceptualization, Data curation, Formal analysis, Investigation, Methodology, Software, Validation, Visualization, Writing - original draft, Writing - review and editing; Leonora S Bittleston, Conceptualization, Data curation, Formal analysis, Methodology; Gabriel E Leventhal, Conceptualization, Data curation, Methodology, Software; Lu Lu, Methodology; Otto X Cordero, Conceptualization, Formal analysis, Funding acquisition, Project administration, Supervision, Writing - review and editing

### Author ORCIDs

Akshit Goyal http://orcid.org/0000-0002-9425-8269
Otto X Cordero http://orcid.org/0000-0002-2695-270X

### Decision letter and Author response

Decision letter https://doi.org/10.7554/eLife.74987.sa1
Author response https://doi.org/10.7554/eLife.74987.sa2

## Additional files

### Supplementary files

• Supplementary file 1. Metadata and accession numbers for all 33 assembled genomes used in the study.

• Supplementary file 2. Set of single-nucleotide polymorphism (SNP) locations and corresponding gene annotations for members of an example community M04.

• Transparent reporting form

### Data availability

Raw sequencing reads are available in the NCBI Sequence Read Archive (BioSample SAMN17005333). Assembled genomes have been deposited in the NCBI GenBank database (BioProject PRJNA682646). Genome metadata and accession numbers are provided in Supplementary file 1.

The following datasets were generated:

| Author(s) | Year | Dataset title | Dataset URL | Database and Identifier |
|---|---|---|---|---|
| Goyal et al | 2021 | Interactions between strains govern the eco-evolutionary dynamics of microbial communities | https://www.ncbi.nlm.nih.gov/biosample/?term=SAMN17005333 | NCBI BioSample, SAMN17005333 |
| Goyal et al | 2021 | Interactions between strains govern the eco-evolutionary dynamics of microbial communities | https://www.ncbi.nlm.nih.gov/bioproject/?term=PRJNA682646 | NCBI BioProject, PRJNA682646 |

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
