## [Editor Report]

How easily is one species replaced by another system in an ecosystem, and what does it take so that two species are no longer equivalent? This is a central issue of ecology, which has been addressed in this elegant study. The rule of thumb the authors come up with, that genetic differences between two bacterial strains greater than about 100 bp are a good predictor of these strains being no longer ecologically equivalent, is likely to be one that will be highly cited in future.

---

## [Decision Letter]

**Decision letter after peer review:**

Thank you for submitting your article "Interactions between strains govern the eco-evolutionary dynamics of microbial communities" for consideration by *eLife*. Your article has been reviewed by 3 peer reviewers, and the evaluation has been overseen by me as Senior and Reviewing Editor. The reviewers have opted to remain anonymous.

A central question in ecology is to what extent individual species can be replaced by other species with similar functional attributes. This study addresses this question using an innovative approach that begins with natural microbial communities harvested from similar niches in nature, bringing these communities to the lab and then letting them evolve under identical conditions. The conclusion is that even seemingly minor genetic differences can have outsized effects. Importantly, the authors come up with a rule of thumb, that genetic differences between two strains greater than about 100 bp are a good predictor of these strains being no longer ecologically equivalent. Because this number is easy to remember, it will likely be broadly cited, and it is therefore important to ensure that this estimate is not too far off the mark.

Essential revisions:

I would like you to address all the comments of the reviewers, but please pay particular attention to the following items:

1. There needs to be a better understanding of what "strain" means. Please explore the data more fully to estimate how many different lineages correspond to single "strains". If you have good arguments that strains are the equivalent of isolates, please state so.

2. Determine whether different strains are likely to have diverged within the host, or whether they constitute different colonization events.

3. State effect sizes when discussing inter-specific versus intra-specific correlations.

4. State how well the statistical approach used can detect positively or negatively coupled oscillations with or without a time-lag between strains, and explore how robust the conclusions are if these patterns remain undetected.

5. State if the correlational analysis would allow the identification of higher-order interactions.

*Reviewer #1 (Recommendations for the authors):*

Species were defined as having identical 16S rRNA (l. 97-98), a practice known as amplicon sequence variants (ASVs). Normally, V4 region sequencing is usually resolved to operational taxonomic units (OTUs) of 97% similarity representing something between the species and general level. There are benefits for either method, and ASVs may be problematic due to dissimilarities between different copies of the 16S rRNA V4 region in the same microbe. Also, different species in certain genera may have similar V4 regions. Since much of the accuracy of their method depends on this species inference, the authors should detail how they resolve species and possibly compare it to alternative methods.

Strains were resolved from SNPs that were highly correlated (mean R=0.8 according to line 144-145). What if this threshold changed? Would there be additional strains? How would that affect the cross correlation between strains within a species? Does the number of reads mapped to a certain genome affect this figure? Figure S5 shows a clear separation between strains (although in both panels one could spot 3 and not 2 strains), but how does this look in other species? If it's not as clear, this requires some benchmarking of clustering thresholds to see if the conclusion still holds.

Some of the results depend on the steady-state of the microbial community. The authors state there is a 31% temporal coefficient of variation (l. 106). Is that a little? A lot? How does it affect the relevant results? Figure S1 shows communities that are more stable than others: while M01 and M08 seem very stable, M02, M04, M06 and M10 seem quite variable. How do the results differ between stable and unstable communities?

All strain-strain correlation histograms and plots show a range of [0,1], but the range of coefficients for the Pearson correlation is [-1,1]. How do the authors explain this discrepancy? I could not understand it from the Methods section. In the case that only the magnitude of correlation (its absolute value) was considered, this is probably wrong, as two strains that have a negative correlation should probably not be considered coupled.

Question:

Does the distribution across genes of the SNPs separating two across genes determine strain dynamics? I.e., do you see clearer separation between strains if the SNPs fall within a certain functional group?

*Reviewer #2 (Recommendations for the authors):*

1. The authors have performed a number of interesting analyses of the genetic basis of strain coexistence. Looking at Figure 2C, it is clear that when there are O(104), strains are nearly always decoupled, whereas strains with <O(102) SNVs between them are nearly always coupled (a fact which the authors comment on). Interestingly, recent studies on the human gut microbiome (Garud, Good et al., 2019 and Zhao, Lieberman et al., 2019) estimated the number of SNVs which segregate between strains of a species found in unrelated human hosts is also O(104), while much smaller numbers of segregating sites (O(102)) accumulate due to clonal diversification within the host. This opens the interesting possibility, which the authors might consider exploring, that “coupled” strains are lineages which diversified within the host plant at some point prior to transplantation to the lab, while “decoupled” strains are strains which independently colonized the host. If the authors are able to confidently reconstruct haplotypes using the strai“ phasin” scheme employed, it should be straightforward to estimate the time to the most recent common ancestor (TMRCA) between coexisting strains, and then compare this number with the lifespan of the plant host species. This analysis would be highly interesting, as it might clarify the timescales over which evolutionary modifications accumulate into meaningful ecological differences between strains.

2. The authors outline an elegant model to determine if strain interactions are stronger than expected under an ecological null model in their Methods section (lines 652 – 686) and then actually use the significance test developed there to show that strains have a denser network of interactions than species in one community (M07), a result which is only shown in Supplementary Figure 15. Why is this analysis relegated to the Supplement? So far as I can tell, this network of interactions analysis is in fact not referenced in the main text, which seems a pity as it is an interesting analysis. Additionally, what do the results of this network of interactions analysis look like in the other 9 communities? If strains typically have a denser network of interactions across communities – if not, does this impact the conclusions of the paper?

3. I personally feel this test of interaction significance is in fact stronger than the relabeling scheme outlined in the main text as it tests for the significance of interactions relative to a well-defined model.

4. To build off the previous point, it strikes me as necessary to actually establish that inter-specific strain correlations are actually meaningfully stronger than species correlations, rather than just stronger. Are 76% of strain couplings significantly stronger than the corresponding species abundance trajectories, or marginally so? It would be helpful if the authors could mention the effect sizes at play here

5. Additionally, I am confused as to why the correlation between species abundance trajectories is not simply a weighted average of the respective strain abundance correlations.

6. I would argue that the model outlined in the Methods, which is used to build the interaction network is not a “neutral” model (line 664), as the authors state. Rather, it is an ecological null model.

7. Could the reason that strain abundances appear more “dynamic” have something to do with the relatively greater strength of sampling noise within vs. between species? I would imagine that there could be greater variance in strain abundances/differences in correlations due to the fact that species abundances/correlations were measured using 16S and strain dynamics were measured with shotgun reads. Why not standardize everything with shotgun reads? Species abundances can be estimated using shotgun reads, as well. This strikes me as a potentially important source of technical noise.

*Reviewer #3 (Recommendations for the authors):*

– Please explain in the main manuscript (i) how sensitive the statistical approach used is to also detect positively or negatively coupled oscillations with or without a time-lag between strains, (ii) how robust the conclusion are if these patterns remain undetected, and (iii) if the correlational analysis would allow to identify also higher-order interactions (i.e. beyond pairwise, including dynamics resulting from >3 way interactions).

– I do not fully understand the importance of pseudogenization for this study. The statement in Line 386 about deactivation of genes that do not contribute to fitness is very general and I would expect this phenomenon to be independent of strain-level differentiation. Therefore, also the conclusion in Line 388, that non-functional pseudogenization contributes to strain-specific interactions is not fully clear. Please clarify.

– I would like to suggest the following (new) title to better reflect the main finding of the study: „Strain-level rather than species-level differences govern eco-evolutionary dynamics in microbial communities”.

– The authors ask what SNPs lead to a decoupling of strains. However, would it also be possible to define a set of genes that lead to coupling? For example, can some metabolic interactions (cross-feeding, competition, etc.) be inferred from strains that showed coupling? This would significantly enhance the manuscript, because it would provide a mechanistic explanation for the observed pattern.

---

## [Author Response]

Essential revisions:I would like you to address all the comments of the reviewers, but please pay particular attention to the following items:1. There needs to be a better understanding of what “strain” means. Please explore the data more fully to estimate how many different lineages correspond to single “strains”. If you have good arguments that strains are the equivalent of isolates, please state so.

We have added new text and analysis to support the notion that each of our strains correspond to distinct lineages, which diverged outside the pitchers from which they were sampled. We have also clarified our usage of the word “strain”, both in the Results and Methods sections, as well as new Supplementary Figures showing that our assignment of strains is robust to the clustering methods we employed.

2. Determine whether different strains are likely to have diverged within the host, or whether they constitute different colonization events.

As suggested, we have now performed this analysis, based on which we believe that the presence of each separate strain corresponds to a different colonization event (Reviewer #2, point 1).

3. State effect sizes when discussing inter-specific versus intra-specific correlations.

We have now calculated and reported the effect sizes of the greater strength of intra-specific correlations in the text (lines 271).

4. State how well the statistical approach used can detect positively or negatively coupled oscillations with or without a time-lag between strains, and explore how robust the conclusions are if these patterns remain undetected.

The revised text now explicitly states our temporal limit of resolution, and argues why our results are robust to only being able to correlations within this resolution (lines 703-713).

5. State if the correlational analysis would allow the identification of higher-order interactions.

We have now discussed that more data would be needed to identify higher-order interactions using our method (lines 709-713).

Reviewer #1 (Recommendations for the authors):Species were defined as having identical 16S rRNA (l. 97-98), a practice known as amplicon sequence variants (ASVs). Normally, V4 region sequencing is usually resolved to operational taxonomic units (OTUs) of 97% similarity representing something between the species and general level. There are benefits for either method, and ASVs may be problematic due to dissimilarities between different copies of the 16S rRNA V4 region in the same microbe. Also, different species in certain genera may have similar V4 regions. Since much of the accuracy of their method depends on this species inference, the authors should detail how they resolve species and possibly compare it to alternative methods.

We understand the reviewer’s concern that there might be a few different ways to identify “species” in our communities: either coarsely-resolved OTUs or finely-resolved ASVs. We have two points to make in this regard, depending on two possible interpretations of their comment.

First: the reviewer is asking us to be more careful and explicit about using the term "species" for identical 16S ribotypes or ASVs, because historically species used to imply OTUs. As suggested, in the revised manuscript, we have included new text clarifying our usage of the term, as well as clearly explaining in detail how we resolved species (lines 142 and 517).

Second: the reviewer is asking us to be more coarse in our resolution by looking at OTUs. However, we believe that for our manuscript, it’s essential to work at the highest level of resolution, since our point is that even at finer resolutions, there are differences in eco-evolutionary dynamics. Even when we resolve dynamics finer than identical 16S ribotypes (ASVs), we see strains with decoupled dynamics. This key takeaway of our work will remain unchanged even if we coarsen the resolution and perform our analysis at the OTU level, which was historically used to delineate species due to the lack of accurate sequencing technology and ASV-resolving algorithms.

Strains were resolved from SNPs that were highly correlated (mean R=0.8 according to line 144-145). What if this threshold changed? Would there be additional strains? How would that affect the cross correlation between strains within a species? Does the number of reads mapped to a certain genome affect this figure? Figure S5 shows a clear separation between strains (although in both panels one could spot 3 and not 2 strains), but how does this look in other species? If it's not as clear, this requires some benchmarking of clustering thresholds to see if the conclusion still holds.

As requested by the reviewer, in the revised manuscript, we have included a new Figure 1 —figure supplement 7 showing more examples of SNP clusters, as well as a histogram of the number of clusters observed across the entire dataset. In what follows, we will respond to the reviewer’s questions in this comment.

First, we wish to clarify that we did not use a threshold correlation value to resolve strains from SNPs. In lines 151 of the original manuscript, we reported the average correlation between the SNPs trajectories of a species, which we stated was quite high (0.8). The value of the average correlation coefficient suggests genetic linkage between SNPs. Importantly, it was not used to calculate the number of strains; for that, we used the clustering approaches outlined in Methods (lines 570-582). Two different approaches suggested that we could not resolve more than two strains for any species in our data, in part due to limitations of sequencing depth, temporal sampling, etc., which we have highlighted in lines 592-595. Thus, we believe that our results regarding strain-strain correlations, given our data, are robust to methodological changes.

Second, Figure 1 —figure supplement 6 shows 2 clusters of SNPs and one set of unclustered SNPs (which do not cluster together by temporal correlation), but as shown in Figure 1 —figure supplement 5 and explained in lines 570-582, the last set of unclustered SNPs always corresponds to a set of ‘nearly fixed’ SNPs, with frequencies close to 1. Using the pigeonhole principle, our interpretation of such a cluster of fixed SNPs is that these SNPs don’t represent a distinct strain; instead they represent mutations in both strains relative to our reference genome (which is why they show up as SNPs), and that these mutations are common to both strains (which is why their frequency is always ~1 in the sample).

Some of the results depend on the steady-state of the microbial community. The authors state there is a 31% temporal coefficient of variation (l. 106). Is that a little? A lot? How does it affect the relevant results? Figure S1 shows communities that are more stable than others: while M01 and M08 seem very stable, M02, M04, M06 and M10 seem quite variable. How do the results differ between stable and unstable communities?

To us, a temporal coefficient of variation of 0.31 implies low abundance fluctuations compared to the mean abundance of each species. When communities are far from steady state (as in the first 3-4 transfers of our experiment, see Bittleston et al., *Nat. Comms.* (2020)), changes in relative abundances are much larger than the mean abundance (temporal coefficient of variation much larger than 1).

Further, we have more evidence which suggests that the communities are near steady-state. Figure 1 —figure supplement 2 shows a low-dimensional representation of the dynamics of all communities at the 16S level, using NMDS. This plot also shows that community compositions do not fluctuate wildly, instead staying within a small region of NMDS space around their respective equilibria.

That being said, we understand the reviewer’s concern that some communities will have more variable dynamics than others. To demonstrate that the variability in dynamics does not systematically affect our main results, we have included a new Figure 3 —figure supplement 2, similar to Figure 3b, which shows a scatter plot of species- and strain-level correlations colored by individual communities. This plot shows that more variable communities (M02 or M10) are not more likely to have higher strain-level correlations than less variable communities (M01 or M08).

All strain-strain correlation histograms and plots show a range of [0,1], but the range of coefficients for the Pearson correlation is [-1,1]. How do the authors explain this discrepancy? I could not understand it from the Methods section. In the case that only the magnitude of correlation (its absolute value) was considered, this is probably wrong, as two strains that have a negative correlation should probably not be considered coupled.

We only plotted the magnitude of strain-strain correlations, but we understand the reviewer’s concern regarding negative correlations. There are two reasons why we plotted the magnitude of correlation coefficients: (1) for simplicity, and (2) because there were very few negative correlations, most of low magnitude (<0.3). The latter were correctly classified as decoupled even under the simpler magnitude-only definition of coupling. In the revised manuscript, we have included the full distribution of strain-strain correlations, with their sign, to demonstrate our point (Figure 2 —figure supplement 3). We have also explicitly discussed it in lines 696-699 of the revised manuscript.

Question:Does the distribution across genes of the SNPs separating two across genes determine strain dynamics? I.e., do you see clearer separation between strains if the SNPs fall within a certain functional group?

This is an intriguing point, one that we had also thought about while originally analyzing the data. Unfortunately we found no such clear functional gene groups that distinguished strains based on their dynamics, outside of the usual gene groups that differentiated coexisting strains, i.e., transporters, regulators and carbon-catabolizing enzymes (Figure 4a).

Reviewer #2 (Recommendations for the authors):1. The authors have performed a number of interesting analyses of the genetic basis of strain coexistence. Looking at Figure 2C, it is clear that when there are O(104), strains are nearly always decoupled, whereas strains with <O(102) SNVs between them are nearly always coupled (a fact which the authors comment on). Interestingly, recent studies on the human gut microbiome (Garud, Good et al., 2019 and Zhao, Lieberman et al., 2019) estimated the number of SNVs which segregate between strains of a species found in unrelated human hosts is also O(104), while much smaller numbers of segregating sites (O(102)) accumulate due to clonal diversification within the host. This opens the interesting possibility, which the authors might consider exploring, that "coupled" strains are lineages which diversified within the host plant at some point prior to transplantation to the lab, while "decoupled" strains are strains which independently colonized the host. If the authors are able to confidently reconstruct haplotypes using the strain phasing scheme employed, it should be straightforward to estimate the time to the most recent common ancestor (TMRCA) between coexisting strains, and then compare this number with the lifespan of the plant host species. This analysis would be highly interesting, as it might clarify the timescales over which evolutionary modifications accumulate into meaningful ecological differences between strains.

This is an excellent and very interesting suggestion, and we would like to thank the reviewer for it. In the revised manuscript, we have reported the results of the suggested analysis (lines 141-144 and lines 677-687). We found that the lifetime of a single pitcher (~3 months) is much shorter than the average fixation time for even a single mutation (~1 year; details of the calculation in Methods, lines 677-687). Therefore, we believe that it is highly unlikely that any pair of strains, which we observed to coexist, diverged during the lifetime of a single pitcher.

Our understanding is thus that each strain pair most likely arrived in the same pitcher via distinct colonization events. They may have diverged in association with other pitchers in the same population (the community persists in a frozen block of ice in a pitcher over winter). From this metapopulation, the strains may have dispersed into the pitchers we sampled (perhaps from older, neighboring pitchers).

2. The authors outline an elegant model to determine if strain interactions are stronger than expected under an ecological null model in their Methods section (lines 652 – 686) and then actually use the significance test developed there to show that strains have a denser network of interactions than species in one community (M07), a result which is only shown in Supplementary Figure 15. Why is this analysis relegated to the Supplement? So far as I can tell, this network of interactions analysis is in fact not referenced in the main text, which seems a pity as it is an interesting analysis. Additionally, what do the results of this network of interactions analysis look like in the other 9 communities? If strains typically have a denser network of interactions across communities – if not, does this impact the conclusions of the paper?

We thank the reviewer for their interest in this analysis. In the revised manuscript, we have referenced and summarized this analysis in the main text (lines 278-289). The reason that we shifted this analysis to the supplement is we were repeatedly suggested to do so by colleagues in previous drafts of the manuscript, since they believed the consumer-resource model made a stronger, more convincing claim about the observed correlations being strain-specific.

With respect to the reviewer’s follow-up question: we have moved the section to Methods, and in it, we state that across all 10 communities, strain interaction networks were denser than species interaction networks, about 70% denser on average (lines 811-819 and lines 273-274 in the main text).

3. I personally feel this test of interaction significance is in fact stronger than the relabeling scheme outlined in the main text as it tests for the significance of interactions relative to a well-defined model.

In our opinion, both tests apply to different questions. The test of interaction significance asks for the presence of a specific interaction (whether a correlation is meaningfully different from zero), while the relabeling scheme asks about the relative magnitude of a pool of interactions. We agree that the relabeling scheme does not contain a well-defined model, but we believe our analysis using a consumer-resource model (Figure 3e-g) remedies this. Specifically, it shows that the relatively higher magnitude of strain-strain correlations compared with species-species correlations is expected only when strains are phenotypically distinct in how rapidly they grow on a variety of resources.

4. To build off the previous point, it strikes me as necessary to actually establish that inter-specific strain correlations are actually meaningfully stronger than species correlations, rather than just stronger. Are 76% of strain couplings significantly stronger than the corresponding species abundance trajectories, or marginally so? It would be helpful if the authors could mention the effect sizes at play here

In the revised manuscript, we have mentioned the effect sizes associated with the correlational analysis in Figure 3b (line 271). Strain-strain correlations are indeed meaningfully stronger than species-species correlations after accounting for their effect sizes, on average being larger by 0.17 ± 0.01 (one sample Student *t*-test, *P* < 10^-3^ for the mean being significantly different from zero).

5. Additionally, I am confused as to why the correlation between species abundance trajectories is not simply a weighted average of the respective strain abundance correlations.

We do not believe the correlation between species abundances can be obtained using only a weighted average, since the underlying strain dynamics can be decomposed in complicated ways. Consider, for instance, two species, both of whose abundance remains constant over time, but each with two strains whose dynamics are completely anticorrelated with each other. One can see through this simple example that no weighted average between the respective strain abundances (which can be quite high if constructed so), would yield a species-species correlation of zero, which is what the correlation between the two constant-abundance species would be.

6. I would argue that the model outlined in the Methods, which is used to build the interaction network is not a "neutral" model (line 664), as the authors state. Rather, it is an ecological null model.

In the original manuscript, we used the term “neutral model” because our model assumed no fitness differences between different strains (fitness represented by population growth rate at equilibrium). The term “ecological null model” is also appropriate and easier to interpret, and as suggested by the reviewer, we have replaced neutral with null in the revised manuscript. We thank the reviewer for this suggestion.

7. Could the reason that strain abundances appear more "dynamic" have something to do with the relatively greater strength of sampling noise within vs. between species? I would imagine that there could be greater variance in strain abundances/differences in correlations due to the fact that species abundances/correlations were measured using 16S and strain dynamics were measured with shotgun reads. Why not standardize everything with shotgun reads? Species abundances can be estimated using shotgun reads, as well. This strikes me as a potentially important source of technical noise.

This is a useful control. As we showed in the original manuscript, species abundance estimates from both 16S and shotgun reads agree with each other, evidenced by a strong positive correlation between them (Figure 1 —figure supplement 10). In the revised manuscript, we have included a new Figure 3 —figure supplement 3, showing our correlational analysis from Figure 3b repeated using species abundance estimated using shotgun reads. This new analysis shows that our key results still hold, i.e., the majority of strain-strain correlations are greater than species-species correlations.

Reviewer #3 (Recommendations for the authors):– Please explain in the main manuscript (i) how sensitive the statistical approach used is to also detect positively or negatively coupled oscillations with or without a time-lag between strains, (ii) how robust the conclusion are if these patterns remain undetected, and (iii) if the correlational analysis would allow to identify also higher-order interactions (i.e. beyond pairwise, including dynamics resulting from >3 way interactions).

Thank you for this suggestion. In the revised manuscript, we have now included text explaining the following:

i) our limit of detection of the time-lag between strain coupling is 37 generations, owing to our temporal resolution; we cannot detect correlations with a time lag less than this [lines 703-707];

ii) the robustness of our conclusions still stands, since the presence of any undetected correlations between strains, with a shorter time lag, would only strengthen our conclusions [lines 707-709];

iii) we do not believe that we have enough data to detect higher-order interactions with our current methods; for that, we will need to sample many more than 10 communities, so that we have plenty of cases where two strains A and B are present both in a community with another strain C, and in one without strain C [lines 709-713].

– I do not fully understand the importance of pseudogenization for this study. The statement in Line 386 about deactivation of genes that do not contribute to fitness is very general and I would expect this phenomenon to be independent of strain-level differentiation. Therefore, also the conclusion in Line 388, that non-functional pseudogenization contributes to strain-specific interactions is not fully clear. Please clarify.

In the revised manuscript, we have clarified how pseudogenization may contribute to strain decoupling (lines 380 and lines 398). Our goal in the section on pseudogenization was to point out that conspecific strains are different from each other not just by mutations, but also by differential pseudogenization. One of the two coexisting strains had a set of genes pseudogenized, while the other didn’t. Thus, decoupled strains might have different dynamics either due to their mutations, or their pseudogenes.

Mechanistically, our rationale was as follows: since the genes that were pseudogenized in our communities had functions that, to us, seemed irrelevant in our experimental conditions, we conjectured that strains with pseudogenes might provide a fitness advantage, in turn leading to dynamical differences, or decoupling.

– I would like to suggest the following (new) title to better reflect the main finding of the study: “Strain-level rather than species-level differences govern eco-evolutionary dynamics in microbial communities".

We thank the reviewer for this suggestion. We agree that the suggestion reflects the key finding of our work. We feel that our current title is shorter and easier to read, and hence we prefer to stick to it.

– The authors ask what SNPs lead to a decoupling of strains. However, would it also be possible to define a set of genes that lead to coupling? For example, can some metabolic interactions (cross-feeding, competition, etc.) be inferred from strains that showed coupling? This would significantly enhance the manuscript, because it would provide a mechanistic explanation for the observed pattern.

This is an interesting suggestion, and as we stated in an earlier response, one that we tried to explore in our original analysis. However, we could not detect statistical differences between gene categories of coupled and decoupled strains, only coexisting from non-coexisting strains (Figure 4 and Figure 4 —figure supplement 3 respectively).